# Seismically imaged lithospheric delamination and its controls on the Mesozoic Magmatic Province in South China

Haijiang Zhang [1,2] ✉, Qing-Tian Lü[3], Xiao-Lei Wang [4], Shoucheng Han [1], Lijun Liu[5], Lei Gao[1,3], Rui Wang [6] & Zeng-Qian Hou[3,7] ✉

The current lithospheric root of the South China Block has been partly removed, yet what mechanisms modified the lithospheric structure remain highly controversial. Here we use a new joint seismic inversion algorithm to image tabular high-velocity anomalies at depths of ~90–150 km in the asthenosphere beneath the convergent belt between the Yangtze and Cathaysia blocks that remain weakly connected with the stable Yangtze lithosphere. Based on obtained seismic images and available geochemical data, we interpret these detached fast anomalies as partially destabilized lower lithosphere that initially delaminated at 180–170 Ma and has relaminated to their original position after warming up in the mantle by now. We conclude that delamination is the most plausible mechanism for the lithospheric modification and the formation of a Mesozoic Basin and Range-style magmatic province in South China by triggering adiabatic upwelling of the asthenosphere and consequent lithospheric extension and extensive melting of the overlying crust.

There are two principal mechanisms for the deep recycling of Earth's materials, i.e., subduction and delamination, by which a significant portion of the lithosphere is recycled into the deeper mantle[1]. Delamination is a density-driven process of foundering by convective removal of the lower crust and/or lithospheric mantle[2]. As a type of vertical and spatially localized tectonics, delamination is difficult to detect with geophysical imaging, due to the small scale of the resulting lithospheric drips within the asthenosphere and the commonly envisaged transient nature of the process[3]. Possible cases of lithospheric delamination are reported below Colorado Plateau[4], Great basin[5], Sierra Nevada region in California[6,7], eastern Carpathians[8], Canadian Cordillera[9], the southern Atlantic[10], and Sulu-Dabie orogens[11], where seismically fast bodies are revealed within the upper mantle. It remains unclear whether similar delamination events have occurred below other parts of East Asia and how they were related to widespread intracontinental tectonic and magmatic processes.

The South China block in east Asia formed by the Neoproterozoic amalgamation of the Yangtze and Cathaysia blocks[12,13] (Fig. 1), and is characterized by the development of the world-class Mesozoic Basin and Range-style magmatic province[14]. Its southeast part is physiographically similar to the Basin and Range Province in the western United States where synextensional magmatism was developed[15]. It underwent the Paleozoic orogeny[16], Triassic collision with the North China block and the Indosinian block[17], and the Late-Mesozoic lithospheric reworking and extensive magmatism[14,18,19]. The current lithosphere of the South China block has been largely modified or removed, as evidenced by significant lithospheric extension and thinning (to present depths of ~70 km[20,21]). The causes of both the Mesozoic Basin-

[1]University of Science and Technology of China, School of Earth and Space Sciences, Hefei, China. [2]Mengcheng National Geophysical Observatory, University of Science and Technology of China, Hefei, Anhui, China. [3]Chinese MNR Laboratory of Deep Earth Sciences and Technology, Chinese Academy of Geological Sciences, Beijing, China. [4]State Key Laboratory for Mineral Deposits Research, School of Earth Sciences and Engineering, Nanjing University, Nanjing, China. [5]Department of Geology, University of Illinois Urbana-Champaign, Urbana-Champaign, USA. [6]State Key Laboratory of Geological Processes and Mineral Resources, CUGB, Beijing, China. [7]Institute of Geology, Chinese Academy of Geological Sciences, Beijing, China. ✉e-mail: zhang11@ustc.edu.cn; houzengqian@126.com

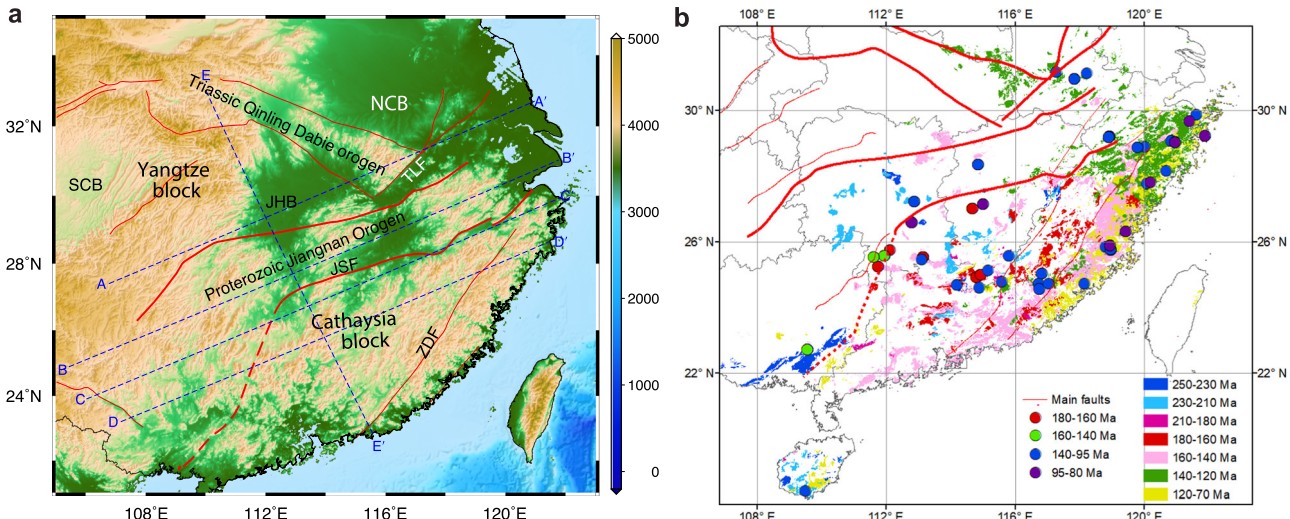

**Fig. 1 | Tectonic framework and Mesozoic magmatic distribution of the South China block. a** Tectonic and topographic map of the South China block. Blue dashed lines mark the position of the tomography imaging profiles (A-A', B-B', C-C', D-D', and E-E'). Solid red lines outline major tectonic units (i.e., the Yangtze block, Cathaysia block, Triassic Qinling-Dabie orogen, and Proterozoic Jiangnan orogen) and major faults (JSF: Jiangshan-Shaoxing fault; TLF: Tanlu fault; ZDF: Zhenghe-

Dabu fault). SCB: Sichuan Basin; JHB: Jianghan basin; NCB: North China block. **b** Spatio-temporal distribution of Mesozoic magmatism in the South China block. The names of tectonic units and faults are the same as **a**. The solid circles show the locations of mafic rocks of different stages, while the colored squares show different stages of granitoids and volcanic counterparts. There is an outward expansion tendency for the distribution of the Late Mesozoic felsic rocks[19].

Range province and the altered present lithospheric structure remain controversial, with previously proposed models including (1) flat subduction and subsequent slab foundering (e.g. Li and Li[14] and Li et al.[22]) or rollback (e.g. Zhou and Li[23] and Dai et al.[24]) of the paleo-Pacific oceanic lithosphere at the convergent margin, and (2) lithospheric delamination (e.g. Wang et al.[25]) and extension (e.g. Deng et al.[21]) in active rifts or intra-continental setting in South China. The subduction model, though a widely-held view, has difficulty in interpreting some key geological and geochemical characteristics of the late Mesozoic igneous rocks in South China[25]. Lithospheric delamination, on the other hand, has been mostly speculated from the perspective of geochemistry, while credible and direct seismic evidence is still lacking.

The existing debates are at least partly due to the poorly resolved upper mantle structure of South China (Gao et al.[26] and references therein) that hampers understanding of the relationship between deep geodynamic processes and surface magmatism. In this study, we developed a high-resolution shear wave velocity (Vs) model for the lithosphere, through joint inversion of seismic body wave arrivals, surface wave dispersion curves, and receiver functions. Our model reveals new structural information about the lithosphere and asthenosphere beneath South China. For the first time, we provide direct seismic evidence for the lithospheric delamination, which we show to have resulted in the formation of the giant Mesozoic Basin-Range province in South China.

## Results and discussion
### Mantle Vp model of South China
To generally understand the velocity structure of the mantle in the study region, we have conducted large-scale mantle seismic tomography in South China (see Methods). The Vp model has a spatial resolution of 1.5° in latitude and longitude and 40–100 km in depth above the mantle transition zone (MTZ), which is determined by teleseismic double-difference tomography[27] by using earthquake arrivals recorded by global stations for events located within China continent and its surrounding areas.

Considerable amounts of P-wave heterogeneities above ~200 km have been observed in South China (Fig. 2a, b). Bounded by the longitude of 110°E, the northwestern part is characterized by a

large-scale high-velocity anomaly down to ~250 km, located beneath the Sichuan basin within the Yangtze block. We interpret this as a residual Archean-Proterozoic lithospheric root that experienced a multiphase reworking (e.g., Li et al.[28]). The southeastern part shows three apparent low-velocity anomalies (LV1, LV2, LV3) with the estimated highest mantle temperatures (1400 °C) at the depths of 120–180 km (Fig. 2b). They roughly occur around the block's convergent boundaries, as LV1 is located beneath the Jiangshan-Shaoxing fault (JSF) between the Cathaysia and Yangtze blocks, LV2 beneath the Tanlu fault (TLF) between the Yangtze and North China blocks, and LV3 at the north edge of the Taiwan island, westward extending to the inland (Fig. 2b), which are also imaged in the global tomography model of Fukao and Obayashi[29] and could be associated with the Okinawa backarc extension. These spatially separated low-velocity anomalies mostly reflect adiabatic upwelling of the asthenosphere among imaged lithospheric discontinuities or vertical weak zones below South China. An EW-trending profile at 28.5°N displays vertical variations in Vp across South China and surrounding regions (Fig. 2c), showing a slightly west-dipping high-Vp Yangtze lithosphere and a low-Vp anomaly to the east (LV3). This profile reveals a significantly different lithospheric structure beneath the Cathaysia block from the stable lithosphere beneath the Yangtze block (Fig. 2c).

### Lithosphere Vs model of South China
To improve the resolution of seismic images for the South China lithosphere, we adopt the newly developed joint inversion method of Han et al.[30] that jointly uses body wave arrival times, surface wave dispersion curves and receiver functions to simultaneously update earthquake locations and constrain three-dimensional P-wave (Vp) and S-wave velocity (Vs) models (see Methods). Due to the complementary sensitivities of the three types of data, this new joint inversion algorithm can better determine both smooth velocity variations and velocity discontinuities. By applying the joint inversion algorithm[30], we have obtained a high-resolution Vs model down to ~150 km in South China (see Methods). Through the checkerboard resolution analysis, the Vs model is well resolved with a spatial grid interval of 0.5° (~50 km) in latitude and longitude and 5-10 km in depth, which can well fit body wave arrival times, surface wave dispersion data, and receiver functions (see Methods). From the Vs model, Vs gradients across the Moho

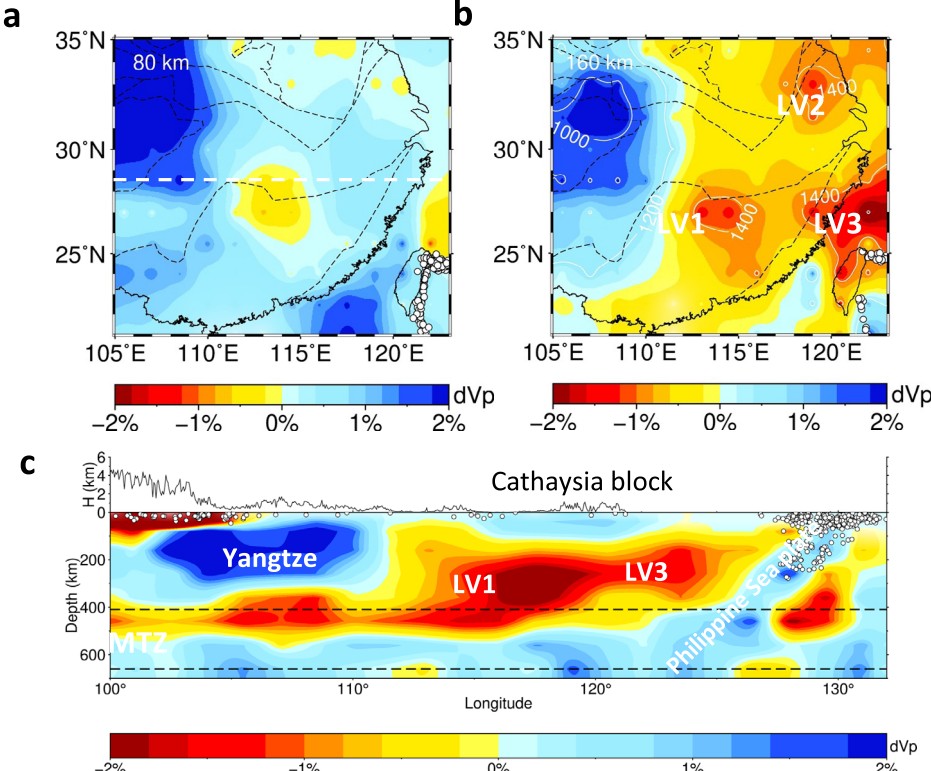

**Fig. 2 | Vp model of South China from mantle seismic tomography. a** Depth slice at 80 km, **b** Depth slice at 160 km, and **c** a cross-section at latitude 28.5° extending from longitude 100° to 132°. The tectonic boundaries in **a**, **b** are the same as Fig. 1. Geological interpretation sees the text. White dashed line in **a** marks the position of latitude of 28.5°. White dots denote earthquakes within 1° of the profile.

can be derived from Vs values at nodes immediately above and below the actual Moho interface, which have the unit of 1/s and can be used to depict the current geodynamic status of the Moho. For a stable Moho, Vs gradients across the Moho are expected to be medium. If the lower crust is delaminated, Vs gradients across the Moho will be large because the velocity differences between the juvenile lower crust and mantle are greater. On the other hand, if the Moho is destructed by the upwelling hot materials, Vs gradients across the Moho would be lower.

Figure 3a–e shows this Vs model of four NE and one NW cross-sections across different tectonic units in South China. Beneath the NE segments along four NE-strike profiles, the lithosphere is generally thinner than ~70 km, consistent with the LAB (lithosphere-asthenosphere boundary) depths determined by receiver functions[20,21]. In contrast, the SW segments are characterized by high-velocity anomalies. Along A-A', the lithosphere beneath the Yangtze block to the west is thick with a deep root, and its northeastern edge shows a vertically stretching low-velocity anomaly upwards to the Moho (Fig. 3a). Along B-B', below ~90 km there are at least three spatially separated high-velocity bodies extending down to ~150 km, which are weakly connected with the overlying lithosphere (Fig. 3b). The velocity features along profile C-C'' are similar to those along profile B-B', but three spatially separated high-velocity bodies beneath ~90 km are located deeper and are completely separated from the overlying lithosphere (Fig. 3c). Along profile D-D' that crosses the Cathaysia block, beneath the southern part of the Cathaysia block (i.e., the Nanling Range granitoid domain[18]), a large T-shaped high-velocity anomaly is isolated in a depth range of ~110–150 km (Fig. 3d). The separated gap between this high velocity and the overlying lithosphere at ~80–110 km is replaced by low-velocity anomalies (LV1 or LV3). A similar configuration has also been shown in E-E' profile, in which an irregular high-velocity body beneath the Cathaysia is located in the mantle at depths >110 km and the Yangtze block still keeps its mantle root (Fig. 3e). The distribution of high Vs anomalies in the mantle lithosphere can also be

seen in the three-dimensional plot (Fig. 4) and different depths (Supplementary Fig. 10) as well as other profiles (Supplementary Fig. 12).

Two significant zones of Vs gradient variation across the Moho have been recognized in South China (Fig. 3f). An arc-shaped low-Vs gradient zone exists around the eastern edge of the Sichuan basin (SCB), and spatially coincides with a dipping zone around 112°E where the Sichuan lithosphere is disconnected from that further east (Fig. 3a). A large NE-trending high-Vs gradient zone is mainly located in west Cathaysia, but extends northeastward across the JSF (Fig. 3f) and spatially coincides with a low-Vp/Vs and shallow-Moho zone in South China (Supplementary Fig. 13).

## Lithospheric modification caused by delamination

Our seismic image reveals that the present lithosphere of South China has been significantly thinned (Fig. 3). The Yangtze block partially preserves most of its ancient mantle root, but the lithosphere beneath most of west Cathaysia has been strongly altered to a highly fragmented configuration. In a 3D view, large volumes of the lower lithosphere below South China surrounding the Sichuan craton are separated from the lithosphere above (Figs. 3c–e and 4), but most of these structures remain laterally connected with the Sichuan lithospheric root at depths greater than ~100 km (Fig. 4). Multiple lithospheric blobs are also weakly connected with the overlying lithosphere along the block convergent belt, i.e., the Jiangnan orogen (Figs. 3b and 4).

The Vp profile at 28.5°N shows that the current Philippine Sea Plate is steeply subducting northwestwards and stagnates within the mantle transition zone (MTZ; Fig. 2c). Some researchers have speculated on a potential causal relationship between the subducted slab and upper-plate structures and tectonics. One hypothesis states that the paleo-Pacific (Izanagi) subduction initiated at ~125 Ma[18,31] and could have triggered asthenospheric upwelling that thermally eroded the overlying lithosphere (e.g., LV3 in Figs. 2 and 3). However, this

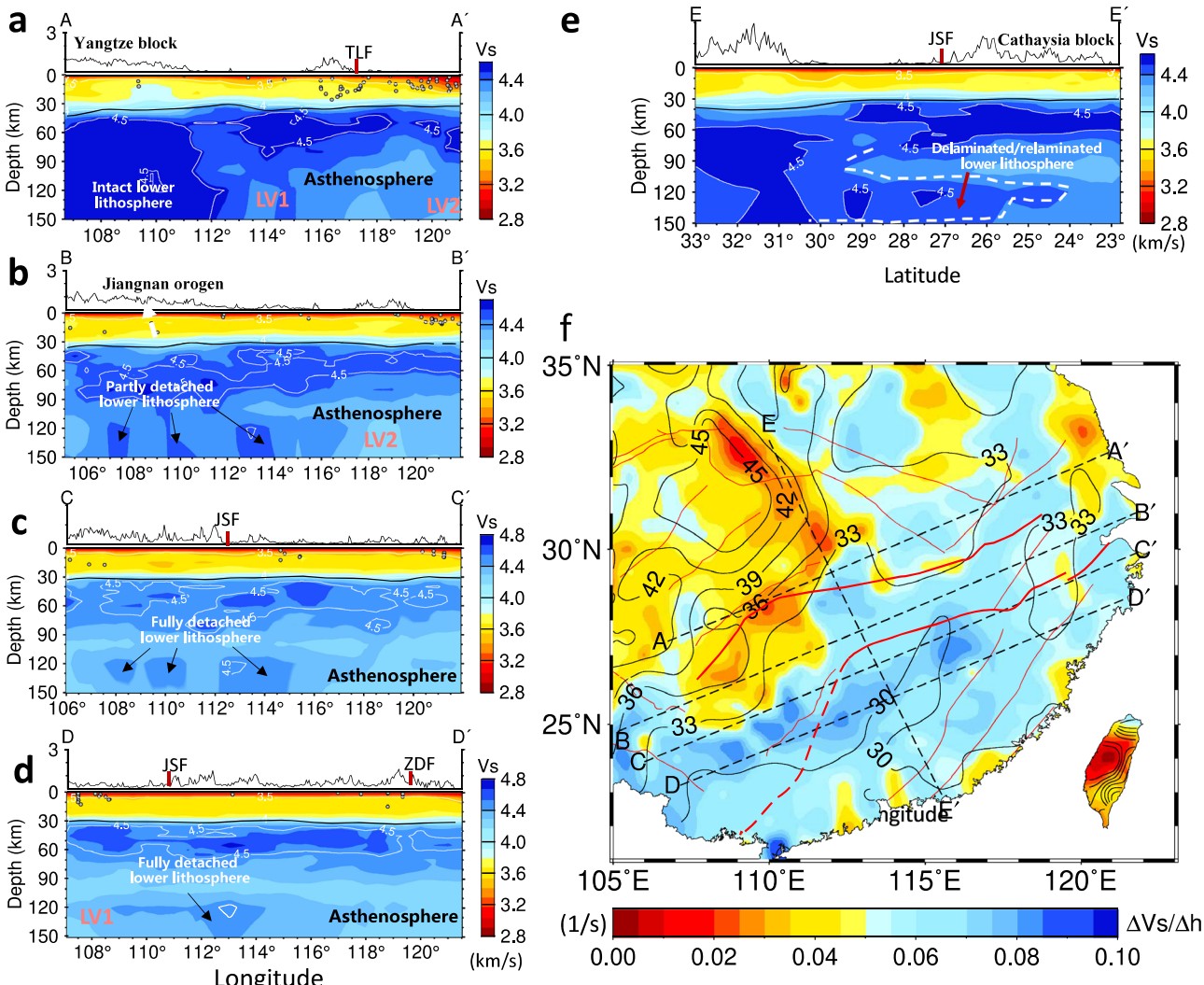

**Fig. 3 | Lithosphere Vs model of South China from joint seismic inversion.** Cross-sections of the Vs model are shown along profiles AA' **a**, BB' **b**, CC' **c**, D-D' **d**, and E-E' **e**, with positions shown in Fig. 1 (also in **f**), respectively. The Vs gradients across the Moho in South China are shown in **f**. The solid black line in cross sections denotes the Moho. See the text for more interpretations.

hypothesis cannot explain the world-class Basin and Ridge-style Mesozoic magmatic province that requires significant lithospheric extension and multipoint-style upwelling of asthenosphere beneath South China (Fig. 2b). Other studies proposed that the paleo-Pacific plate subducted flatly beneath the South China block during the Triassic[14,18,19], an event preceding the ca. 180–170 Ma Jurassic magmatic activities composed of mafic rocks, potassic syenites and A-type granites predominantly developed within the Cathaysia block[25] and accompanying a strong lithospheric extension regime. This model provides a potential connection between Mesozoic magmatism in South China and oceanic subduction. However, the increasing amount of geochronological data does not show an evident trend of westward younging of early Mesozoic magmatism as a flat-slab model implies[19]. In addition, regional subsidence at 180-170 Ma and rapid uplift at ca. 160 Ma observed in South China have been interpreted as the far-field effect of the paleo-Pacific subduction[32], but such fast topographic changes and crustal deformation can be also attributed to lithospheric delamination[33].

We suggest that the Mesozoic tectonism in South China may reflect a composite effect of multiple geodynamic processes including abnormal subduction and lithospheric delamination, and that the Jurassic crustal extension and magmatism mainly resulted from the latter. This delamination proposal is consistent with two major observations

in South China. First, the Mesozoic Basin-and-Range-style magmatic province is dominated by widespread voluminous felsic and granitoid rocks with minor mafic rocks[19] (Fig. 1b), which require an enormous amount of heat and water from the underlying convective mantle[34,35] to trigger extensive melting of the overlying lithosphere, especially the middle to lower crust. Similar cases for delamination have also been proposed in other areas, e.g., the Puna region in the central Andes[36,37] and the southern Sierra Nevada in California[38,39]. Second, the relatively minor mafic rocks in west Cathaysia and Jiangnan orogen show two distinct mantle sources, and yielded a temporal source transition from "older" coexistence of depleted convective mantle and enriched lithospheric mantle at 180–160 Ma to "younger" dominance of enriched lithospheric mantle at 160–95 Ma and to the final coexistence of these two sources at 95–80 Ma (Fig. 5a, b). This source transition records the limited melting of the upwelling asthenosphere at the beginning (i.e., 180–160 Ma), followed by a progressive increase of partial melting of the lithospheric mantle from 160–140 Ma to 140–95 Ma (Fig. 5a, b), a scenario characteristic for lithospheric delamination. The reappearance of depleted mantle melting at 95–80 Ma was evident in the western Cathaysia block (Fig. 5a, b). It was also partly located at the present coastal area (Fig. 1b) and is consistent with the Cenozoic basalts (Fig. 5a, b), which may indicate the effect of paleo-Pacific subduction at the latest Mesozoic[18,40].

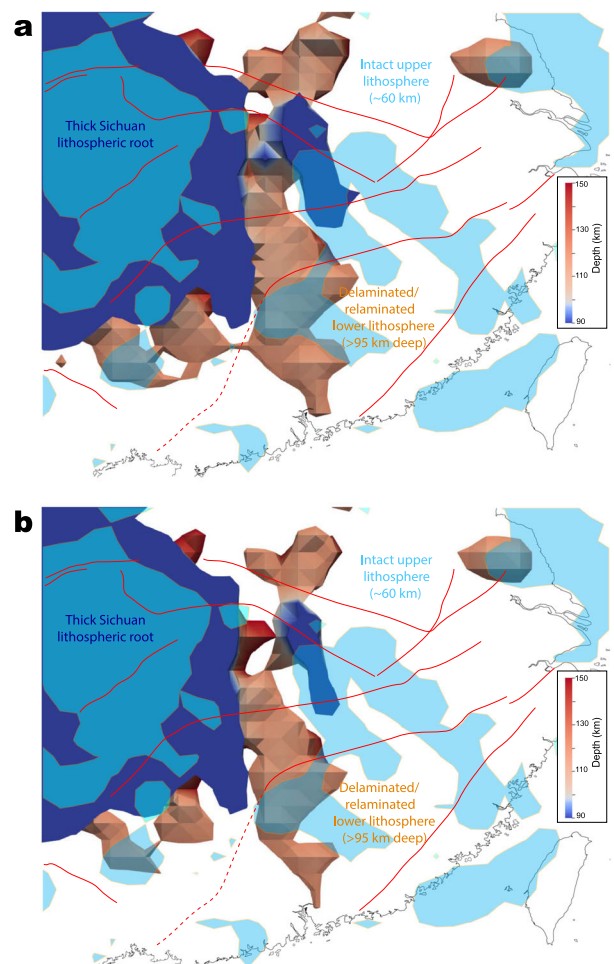

**Fig. 4 | Three-dimensional view of high Vs anomalies in South China mantle lithosphere.** Both Vs anomalies >0.5% **a** and >1.0% **b** are shown for comparison. Vs anomalies refer to Vs perturbations with respect to average velocities at each depth. Colors denote depths where high Vs anomalies are located.

## Mechanism and evolution of lithospheric delamination

The combination of geophysics and geochemistry can provide new insight into the tectonic evolution of South China. The above-mentioned source transition of late Mesozoic mafic magmas in South China (Fig. 5b) suggests that initial delamination likely started at 180–160 Ma. This is not to negate possible slab foundering as suggested earlier[14,41] but to emphasize the linkage between the surface tectonic records and the observed detached high-Vs bodies in the depth range of 90–150 km below South China (Figs. 3 and 4). Following this line of reasoning, the geological observations can further constrain the mechanism for the observed lithospheric delamination in South China. Prior to the Jurassic extension in the South China block, widespread lithospheric shortening took place in Neoproterozoic (the Jiangnan orogeny[13]), Early-Paleozoic (Wuyi-Yunkai orogeny[13]) and Triassic (intracontinental orogeny due to the far-field stress propagation between the Qinling-Dabie orogeny and Indosinian orogeny[17,42]). This multiphase history of crustal thickening and lithospheric shortening should have primed the development of 180–160 Ma mafic rocks and coeval A-type granitoids and potassic syenites in west Cathaysia[18,25]. The continental roots beneath both the Jiangnan and Dabie orogens partly remain at the present (Fig. 3b, e), supporting that orogenic collapse may have triggered the delamination.

Besides lithospheric thickening, the possible existence of an antecedent flat slab beneath south China could further facilitate our identified Mesozoic delamination and the associated tectonic activities.

According to some previous studies, the flat slab foundered[14,22] or rolled back (e.g. Zhou and Li[23] and Dai et al.[24]) during the latest Triassic to earliest Jurassic. Numerical models showed that a flat slab could significantly deform and weaken the overriding lithosphere[24,40], thus creating a necessary condition for subsequent delamination[3,10,43]. The progressive amount of lithospheric loss from the pan-Sichuan region to the southeast coast (Fig. 4) supports the role of this preceding flat slab that has caused more damage in the overriding lithosphere closer to the trench[14,40].

In practice, the evolution of delamination should be controlled by both the lithospheric mechanical state and the underlying mantle dynamics. The magmatic history demonstrates that the associated lithospheric delamination successively expanded northeastwards since ~180 Ma, obliquely crossed the JSF, and finally reached the northeastern Yangtze block at ca. 140–120 Ma (Fig. 1b). Seismic images show that a high-Vs gradient zone across the Moho continuously extends along the JSF northeastwards to reach the TLF (Fig. 3f), which spatially coincides with a low Vp/Vs zone (1.67–1.72[44]) and a thin crust zone (33–29 km[45]) (Supplementary Figs. 13a, b). In particular, at the TLF, where a high-Vs gradient-domain (Fig. 3f) spatially coincides with the LV2 (Fig. 2b), the LAB is the shallowest (~70 km[20]) and the crust is the thinnest (25–29 km[45]). All these data suggest a loss of the lowermost mafic crust along the delamination zone. Geological observations indicate that the crust-derived Mesozoic adakitic rocks emplaced along the margins of the estimated delaminated zone (Fig. 5a, b). These adakitic rocks yielded high MgO and Cr contents, suggesting the derivation from remelting of eclogized mafic lower crust during its delamination into the hot mantle[46,47]. It should be noted that only the locally thickened crustal root would delaminate, a process that can entrain only part of the mantle lithosphere underneath, with other parts of the mantle lithosphere remaining intact.

Although the present lithospheric structures, especially the cold and strong parts, could largely reflect the state after Mesozoic deformation, features that are decoupled from the intact lithosphere, such as the detached lower lithosphere below South China (below>95 km depth in Fig. 4), should be subject to change over time. Our seismic images reveal that the top of the delaminated lithosphere lies approximately at ~90–120 km depth, which seems too shallow and unrealistic for delamination at 180–160 Ma. Recent geodynamic simulations suggest that the delaminated lithospheric segments could regain buoyancy and eventually relaminate to the base of the overlying lithosphere within 100–300 Myr[43]. The observation that the geographic region of the Mesozoic magmatic province (Fig. 1b) is significantly broader than that of the presently imaged detached lower lithosphere (Fig. 4) implies that most of the delaminated lithosphere materials could have been lost into the deep mantle. The preserved portions of the lower lithosphere are either laterally connected with the thick Sichuan cratonic root on the NW or vertically with the upper lithosphere at some locations (Fig. 4). The mechanical coupling among these features should have kept the Mesozoic delaminated lower lithosphere from drifting away during subsequent relamination to their original locations[43].

## Delamination control on the Mesozoic Basin and Range-style magmatic province

Previous models for the Mesozoic magmatic province in the South China Block typically involve a continental arc[48], formed by the Triassic-Jurassic flat subduction[14] and Early Cretaceous slab retreat[18] of the paleo-Pacific plate. Every tectonic model has to explain: (1) our newly-recognized spatial-temporal patterns where magmatism started in west Cathaysia at ca. 180 Ma and gradually migrated outwards till ca. 140 Ma, magmatism during 140–120 Ma forms three juxtaposed volcano-plutonic belts with associated fault-bounded basins, and the 120–70 Ma magmatism shrinks towards the southeast coast[18,19] (Figs. 1b), and (2) a tectono-magmatic transition from a 180–160 Ma

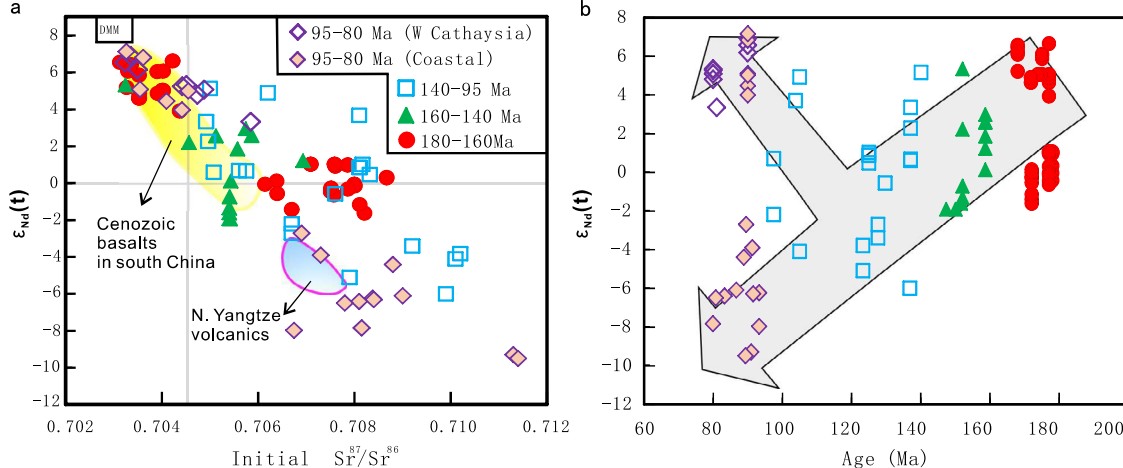

**Fig. 5 | Variation in Sr-Nd isotopes of the late-Mesozoic mafic rocks in west Cathaysia and Jiangnan orogen.** The 95–80 Ma mafic rocks along the present coastal area of South China are also plotted for comparison. Sampling locations can be seen in Fig. 1b. **a** $\varepsilon_{Nd}(t)$-initial $^{87}Sr/^{86}Sr$ diagram showing the secular change of Sr-Nd isotopes of the late Mesozoic mafic rocks. **b** $\varepsilon_{Nd}(t)$-age variations of the same rocks. The 180–160 Ma mafic rocks occurred along the block convergent boundary (CZF-JSF) and within the Cathaysia block. The areas of DMM (depleted MORB mantle) and N. Yangtze volcanics are from Wang et al.[25] for comparison. CZF Chenzhou-Linwu Fault, JSF Jiangshan-Shaoxing Fault.

bimodal suite in a lithospheric extension regime[18,49] to a 160–120 Ma felsic-dominated non-arc suite in a Basin-Range tectonic system[14].

The joint analysis of our high-resolution tomography images (Figs. 2–4) and geological observations (Figs. 1b and 5) outlines a consistent picture (Fig. 6) of Mesozoic lithospheric delamination and resulting decompression melting below the South China Block. The delamination process started below the Jiangnan orogen during early Jurassic, likely triggered by earlier lithospheric thickening and flat subduction, and subsequently expanded outwards, with the final stage of delamination terminated in the NE during early Cretaceous (Fig. 6a). The dominant effect of lithospheric delamination within most of these regions is supported by multiple observations: (1) the widespread mafic eruption (Fig. 1b) with a clear lithospheric signature requires melting at shallow depth, implying in-situ lithospheric thinning, (2) the present-day lithosphere beneath most of South China is very thin (~70 km), and the mostly likely period the required thinning could have happened is the Mesozoic, prior to which was tectonic thickening during early Paleozoic[13,14], (3) Mesozoic mantle xenoliths from the central-south part of the Cathaysia block indicate syn-magmatic sub-continental lithospheric thinning to ~80 km at ca. 170 Ma[50], (4) recent plate reconstructions[51] and present crustal thickness (Supplementary Fig. 13) suggest little lateral extension within South China since the Triassic, so the present-day thin lithosphere is better explained by vertical thinning (delamination) than horizontal stretching, and (5) the presence of large Vs gradients across the Moho indicates wide-spread lithosphere delamination below South China (Fig. 3f).

We suggest that the post-120 Ma magmatism could have resulted from asthenosphere upwelling due to the rollback of the earlier flat paleo-Pacific slab. The peel-off style of the lower-lithosphere delamination and associated upwelling hot asthenosphere (Fig. 6a) exert significant extension on the overlying lithosphere, leading to the Basin and Range-style crustal deformation and magmatism. This delamination style is further reflected in the present-day lithosphere configuration (Figs. 3, 4 and 6b). We argue that the initial delamination at 180–160 Ma along the convergent boundary between the Yangtze and Cathaysia blocks produced a limited scale of lower-lithosphere drips, where both the large depth and the weak asthenospheric upwelling prohibited extensive melting[37], thus generating small-volume mafic magmas. Subsequently thinned and fractured lithosphere allow more efficient adiabatic upwelling that provided enough driving force for lithospheric extension and extensive melting of the overlying crust, forging a NE-trending Basin-Range-style Late-Mesozoic magmatic province.

## Methods

### Mantle seismic tomography of South China

To image the mantle structure of South China, we adopted the tele-seismic double-difference seismic tomography method (teletomoDD) of Pesicek et al.[27], which is a global version of double-difference seismic tomography method (tomoDD)[52,53]. It can use both regional and tele-seismic stations to invert the velocity structure of the study region. We mainly assembled two sets of data from the ISC-EHB Bulletin and China Digital Seismic Network (CDSN) catalog (Supplementary Figs. 1 and 2). The ISC-EHB Bulletin contains a refined version of ISC Bulletin by using the EHB algorithm[54] to minimize errors in earthquake locations. We have collected P-wave arrival times of globally distributed stations in the period of 1964–2017. For the CDSN catalog, we assembled about 780,000 events recorded by 1434 stations during the period of 2008/10 to 2018/06. Most of the events are shallow and of small magnitude. In this study, we carefully selected a limited number of events based on the following criteria. We first divided the study region into $0.5° \times 0.5° \times 50$ km cells and only selected the first 10 events with maximum observations in each cell, and each event has more than 20 observations. From the two data sets, we constructed 7.2 million event-pair differential arrival times by selecting event pairs having more than 50 common observations for the ISC-EHB Bulletin and more than 20 common observations for the CDSN, and the distances between event pairs are between 50 and 500 km.

The model is parameterized by nested grids with a coarse global grid outside the study region and finer regional grid for the study region. For the initial regional model, we choose the AK135 model[55]. Our study region is from 74°E to 136°E in the longitude, from 18°N to 54°N in the latitude, and extends to 2500 km in the depth. The regional grid interval is 1.5° along longitude and latitude, and about 40–300 km in the depth. For the global model, we choose the TX2019slab model[56] as the initial model, with a grid interval of 5° in the horizontal direction and about 55–225 km in depth.

In the inversion, we applied a hierarchy strategy to invert for the velocity structure[52] by first giving absolute arrival times a larger weight and then giving event-pair differential times a larger weight. After 7 iterations, the data root mean square (RMS) residual decreases from 2.255 s to 0.805 s. The checkerboard resolution test was used to check

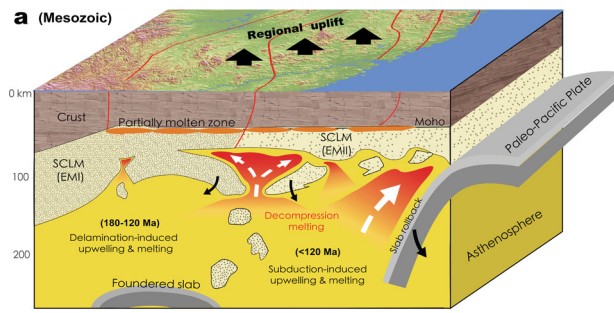

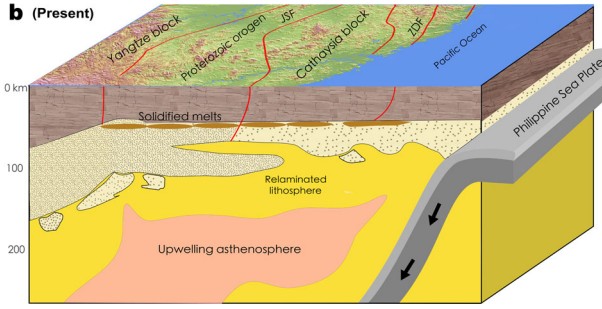

**Fig. 6 | Illustration of South China tectonic evolution since the Jurassic. a** Late Mesozoic (180–70 Ma) lithosphere delamination, asthenosphere upwelling and paleo-Pacific subduction in South China. The asthenosphere upwelling and decompression melting during 180–120 Ma were triggered by the lithospheric delamination, whereas those after 120 Ma were caused by the rollback of the paleo-Pacific subducting slab. **b** The restored lithospheric structure after relamination as imaged today. SCLM subcontinental lithosphere mantle, EMI enriched mantle type one, EMI enriched mantle type two.

the model resolution. In our checkerboard test, we added ±5% anomalies at the adjacent grid nodes in the regional model (Supplementary Fig. 3). Supplementary Fig. 4 shows the checkerboard recovery at different depths. Overall, for South China, the model has a good resolution in most parts.

### Joint seismic inversion of lithospheric structure of South China

To determine high-resolution lithospheric structure of South China, we have adopted a newly developed joint inversion algorithm of seismic body wave arrival times, surface wave dispersion curves and teleseismic receiver functions[30]. Due to the complementary strengths of different seismic data types, the lithospheric Vs model can be more reliably and finely determined, especially in the depth direction[30].

For body wave arrival times, we collected 617,143 P- and 583,628 S-wave first arrivals recorded on 572 seismic stations from 40,028 events over ten years from October 2008 to June 2018 (Supplementary Fig. 5), from which we constructed 4,042,313 P- and 4,049,549 S-wave differential travel times. For surface wave data, the Rayleigh wave phase and group velocity maps with periods ranging from 8 s to 70 s (Supplementary Fig. 6) are extracted from the surface wave tomography of Shen et al.[57].

For the P-wave receiver functions, we used both stations from He et al.[58] and those available from the Incorporated Research Institutions for Seismology Data Manage Center (IRIS DMC). For stations from IRIS DMC, seismic events (Mb greater than 5.5) occurring at epicentral distances between 30˚ and 100˚ were selected. Receiver functions were computed using a time-domain iterative deconvolution method with a Gaussian width of 1.0[59]. Problematic receiver functions were removed using the criteria similar to Chai et al.[60]. The accepted receiver functions were averaged at each station to increase the signal-to-noise ratio. To reduce the scattering noise, we applied the receiver function smoothing/interpolation technique[60] to these single-station-averaged receiver functions. Distance-derived weights were incorporated in the

smoothing/interpolation. The distance-derived weights for stations within a distance of 50 kilometers are set as one to have similar spatial resolutions as surface-wave dispersion measurements. For stations located between 50 and 100 kilometers, the smoothing weight decreases from one to zero as the distance increases. Supplementary Fig. 7 shows the smoothed/interpolated receiver functions used for joint inversion.

Chen[61] determined the Vs model of continental China by inverting the surface wave dispersion curves from Shen et al.[57] by the conventional surface wave tomography. From the Vs model of Chen[61], we constructed a 1-D Vs model for South China by averaging 3D Vs model in the region. 1-D Vp model is then constructed from the 1-D Vs model via the Vp-Vs relationship of Brocher[62]. Then the 1-D Vp and Vs models were extended to a series of 3D grid nodes in the horizontal plane with an interval of 0.5° in both latitudinal and longitudinal directions. In the vertical direction, the grid interval varies from 5 km (shallower than 80 km), to 20 km (80–120 km depth), and to 30 km (120–150 km depth), respectively. For seismic events, the initial earthquake hypocenter parameters were from the earthquake catalog of the China Digital Seismic Network.

Then we conducted joint inversion that utilizes three different datasets by using the algorithm of Han et al.[30]. When performing the joint inversion, we chose the weighting parameters via the trade-off analysis. The weighting parameters for surface wave data and receiver functions were 40 and 45, respectively. When conducting joint inversion with three different seismic datasets, a two-step strategy was taken. The joint inversion of body wave and surface wave data was performed first to get smooth intermediate 3-D velocity models and then receiver functions were incorporated to resolve fine structures related to velocity contrast across the discontinuities. After 12 iterations of joint inversion, the RMS residuals of seismic data are decreased from 1.26 s to 0.50 s, from 0.15 km/s to 0.031 km/s, and from $0.037\,s^{-1}$ to $0.014\,s^{-1}$ for body wave travel times data, surface wave dispersion data and receiver functions, respectively (Supplementary Fig. 8).

To show the reliability of our joint inversion result, we conducted the checkerboard resolution tests[63]. Firstly, the initial models were perturbated alternatively with ±5% anomalies to generate the true checkerboard model (with a horizontal length of 0.5° and vertical length of 5 km at depths shallower than 80 km, 20 km in deeper depths), based on which we calculated the synthetic data. To take the noise in the observed seismic data into consideration, Gaussian random noise (with standard deviations of 0.1 s for body wave travel times, 0.03 km/s for surface wave dispersion curves, and $0.012\,s^{-1}$ for receiver functions) was added to each type of data. Then we performed joint inversion using the same initial models and weighting parameters. The checkerboard patterns and amplitudes are well recovered for Vs model at most depths (Supplementary Fig. 9). The depth slices of the recovered Vs model are shown in Supplementary Fig. 10.

## Data availability

Vs model and geochemistry data generated and assembled in this study have been deposited in the OSF storage accessible at https://doi.org/10.17605/OSF.IO/3PXRK or https://osf.io/3pxrk/

## Code availability

The seismic tomography software package used in this study is available upon request.

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

## Acknowledgements

This research is supported by the National Natural Science Foundation of China under grants 92162322 to X.W., 42230101 & U1839205 to H.Z. and by the Fundamental Research Funds for the Central Universities under grant WK2080000 to H.Z.

## Author contributions

Z.Q.H. and H.Z. designed and initiated the research, interpreted data, and wrote most of the first draft of the manuscript. Q.L. assembled the receiver functions analysis results. X.L.W. assembled and analyzed geochemical data. S.H. conducted the joint inversion and L.G. con-ducted the mantle seismic tomography. L.L., X.L.W. Q.L., and R.W. contributed to the interpretation of results and the revision of the manuscript.

## Competing interests

The authors declare no competing interests
