## [Peer Review File · Nature Communications]

Seismically imaged lithospheric delamination and its controls
on the Mesozoic Magmatic Province in South ChinaREVIEWER COMMENTS

Reviewer #1 (Remarks to the Author):

Whether the partial removal of the deep lithospheric root in South China is caused by subduction or delamination remains a significant controversy. The authors, using a newly developed joint inversion algorithm of seismic body wave arrival times, surface wave dispersion curves, and teleseismic receiver functions, clearly imaged the structural features of the lithosphere-asthenosphere mantle in the lower part of the South China region. The study found that a series of the delaminated lithosphere beneath the convergent belt between the Yangtze and Cathaysia blocks provides very significant evidence for further study of the lithospheric thinning process in south China (even for other ancient Craton). The research will be of interest to the geological and geophysical community. However, there are still some problems that need to be addressed. I therefore suggest publication in NC after minor revision.

Here are my comments.

Comment.1) Lines 26-27: The authors suggest that the lithospheric delamination of the South China Block resulted from the eclogitized mafic lower crust. However, An important condition for eclogitization in the lower crust is lithosphere thickening during orogenesis. The whole South China orogenic event (or lithosphere thickening event) can only be traced back to the Meso-Neo proterozoic, Devonian and Triassic collisional processes between the Cathaysia and Yangtze blocks, while the effects of regional flat-slab subduction are ignored(Li Z X and Li X H, 2007 in *Geology*; Lijun Liu et al., 2021 in *ESR*). Therefore, it raises the question of why the lower crust of the thickening lithosphere during the Meso-Neo proterozoic, Devonian and Triassic collisions suddenly experienced eclogitization until 180-160 Ma and further led to the lithospheric delamination. Please give the reasonable mechanism that triggered lithospheric delamination due to eclogitization of 180-160 Ma.

Comment.2) Lines 56-58: The authors suggest that two existing mechanisms for lithospheric thinning in South China are contradictory, i.g. (1) the flat subduction and slab foundering and (2) the lithospheric delamination. However, the two views may not be contradictory in my opinion. They are two stages of one continuous process. Because slab foundering and lithospheric delamination may correspond to different stage of the lithospheric thinning processes. The migration pattern of magma, tectonic deformation, and metamorphic data together reveal that the initiation of flat subduction may have occurred between 260-250 Ma, while slab foundering occurred at 190-180 Ma, leading to widespread extension of the South China Block. This model matches well all present-day geological observations (including those mentioned in this manuscript) (Li and Li, 2007; Li S.Z., et al, 2019). This model has also been validated by geodynamic numerical simulations. Dai (2020) used dynamical numerical simulations to well validate the possibility of the slab foundering after the flat subduction. Meanwhile, Flat-slab subduction maybe provides two key kinetic conditions for the delamination of the South China lithosphere: (1) the flat-slab subduction process can lead to significant lithospheric thickening of the South China Block (Dai et al., 2020), providing temperature-pressure conditions for the eclogitization of the lower crust, which in turn triggers the delamination of the lithospheric mantle; (2) the flat-slab subduction process can lead to significant thinning and disruption of the lithospheric mantle of the overlying plate (Axen et al., 2018), which may provide favorable conditions for the continued delamination of the overlying slab lithosphere after flat-slab subduction and slab foundering.

Reference:

- [1] Li, Z.X., & Li, X.H. (2007). Formation of the 1300-km-wide intracontinental orogen and postorogenic magmatic province in mesozoic south china: a flat-slab subduction model. *Geology*, 35(2), 179-182.
- [2] S.Z. Li, X.Y Li, J. Zhou, H.H. Cao, et al.(2019). Mesozoic tectono-magmatic response in the East Asian ocean-continent connection zone to subduction of the Paleo-Pacific Plate. *Earth-Science Reviews* 192
- [3] Dai, L., Wang, L., Lou, D., Li, Z., & Yu, S., (2020). Slab rollback versus delamination: contrasting fates of flat-slab subduction and implications for south china evolution in the Mesozoic. *Journal of Geophysical Research- Solid Earth*, 125(4).

[4] Axen, G.J., van Wijk, J.W. & Currie, C.A. (2018). Basal continental mantle lithosphere displaced by flat-slab subduction. *Nature Geosci* 11, 961–964.

Comment.3) Lines 240-244: The manuscript shows the critical tomographic evidence of lithospheric mantle delamination in South China, but it has not provided powerful evidence that the delamination must have started at 180-160 Ma. On the contrary, in Fig. 3 B-B', does the delaminated lithosphere at 120 km depth mean that the South China lithospheric mantle is still undergoing delamination? Besides, in Fig. 1b, there are two mafic rock data located near the JHP Fault (95-80 Ma) (in the manuscript, the authors do not explain the genesis of these two data), which happen to be located above the lithospheric delamination of the South China orogenic belt, does this mean that the beginning of delamination can be fixed at 95-80 Ma? Therefore, more evidence is still needed to verify the timing of this delamination event.

Comment.4) In the C-C' of Fig. 3, the detached lithospheric mantle is completely decoupled from the original lithospheric mantle, while in the B-B' is a weak connection. I suggest the authors can provide a tomographic image of the middle position of these two sections. If the continuous detachment process can be identified, it can well support the authors' important conclusion about the extension of detachment from SW to NE.

Comment.5) In the manuscript, the authors suggest that the lithospheric mantle less than 70km depth has not been completely detached, which resulted from the ecologized mafic lower crust. However, if the ecologized mafic lower crust causes the delamination, the whole lithospheric mantle should be detached into the asthenosphere mantle, not only the lithospheric mantle deeper than 70 km. Therefore, it is controversial that the lithospheric delamination of South China resulted from the ecologized mafic lower crust.

Comment.6) line 53: Wrong format,two periods repeat

Comment.7) Figure 1 should be improved, there are many detailed geological or tectonic maps of the South China Block and its neighbors, I prefer to recommend Sanzhong Li et al.'s paper on South China Block for you.

Reviewer #2 (Remarks to the Author):

NCOMMS-22-36882A-Z

In manuscript titled "Lithospheric delamination evidenced from seismic imaging and its controls on the Mesozoic Basin and Range Magmatic Province in south China", Zhang and colleagues present a new seismic tomography of the lithosphere and asthenosphere under South China and report a group of newly identified high-velocity anomalies that are either dangling at the base of the thin lithosphere or floating in the asthenosphere. They further assess two competing hypotheses, flat-slab subduction vs. delamination, using the spatial distribution and geochemical signature of the Mesozoic igneous rocks in South China. Their favored interpretation is delamination, and they advocate that the magmatism and lithospheric modification in South China imitate the processes that took place in the Basin and Range Province of the western US.

The new tomography derived from the new joint inversion method of Han et al 2022 is a well-thought, rigorous, and state-of-the-art model. The authors have successfully discovered some fine details in the lithosphere and asthenosphere. I therefore believe that this model is worthy of publication. However, the geological/tectonic interpretation is relatively weak, with many loose ends. My concerns are listed below. The first two are the major ones.

My recommendation is Reject, but encourage to resubmit. I would be happy to review this manuscript again.

1. On the evaluation of the flat-slab subduction model:

In Line 188-192, the authors wrote that (1) Li and Li 2007 proposed that flat-slab subduction took place before 170 Ma, and (2) the igneous rocks emplaced between 180-170 Ma suggest an extensional environment. Based on the temporal relationship, the authors ruled out the flat-slab subduction model.

Well, the statement of ">170 Ma" (Line 189) is not wrong, but I am afraid this is rather

misleading. In their Figure 4, Li and Li (2007) explicitly showed that the flat-slab subduction peaked at 210 Ma (or around 200 Ma if you use the age-distance plot in their Figure 3). In their model, the flat slab began to break into pieces and fold "like a taco" (as per Eugene Humphreys in his Farallon flat-slab papers and talks) and sink into the convective mantle by 190 Ma. Please note that ">170 Ma" and "210-200 Ma" are two different concepts in this context. So, just like the Oligocene (~30 Ma) volcanism in the western US cannot be used to rule out the Farallon flat-slab subduction that peaked at ~50 Ma (see, for example, Copeland et al 2017 *Geology* <https://doi.org/10.1130/G38810.1>), the 180-170 Ma igneous rocks in South China cannot be used to rule out the "Paleo-Pacific" flat-slab subduction that peaked at ~210-200 Ma.

I want to clarify that I am not forcing the authors to support the flat-slab subduction model. Instead, my goal here is to help the authors develop an unbiased evaluation of a model. Otherwise, if this is published, anybody could easily write a comment to attack this flaw.

2. On the evaluation of the delamination model:

Let's put the flat-slab subduction model aside and say that the delamination model is correct. But three aspects still concern me.

(1) What caused the delamination, and is it plausible?

The authors suggested that the delamination started at 180-160 Ma and expanded from SW to NE, due to orogenic (or "post-orogenic"?) collapse. Since the most recent orogeny by 180 Ma was the Triassic Qinling-Dabie in the north, why did the delamination begin in the south, more than 500 km away from the Qinling-Dabie orogenic belt? Did the delaminations under Sierra Nevada, Colorado Plateau, or Canadian Rocky Mountains share a similar feature? Are there any numerical modeling studies that suggest such a "far-field effect of delamination"? I am not aware of such things. If there is any, please point it out in the manuscript.

(2) The complexities and uncertainties in delamination and relamination:

The tomography model shows the present-day mantle structure, and the authors focus on the Mesozoic tectonics/magmatism in the discussion. The authors also admitted (Line 240-244) that the delaminated body is too shallow and too long-lived in their favored model. Then they cited Peng et al 2022 *EPSL* to argue that this delaminated body has (or could have) regained buoyancy and could relaminate to the overlying lithosphere. I have three sub-questions regarding these arguments:

First, if a delaminated body has regained buoyancy and been heated by the ambient asthenospheric mantle for >100 Myr, will it still show as a high-velocity anomaly? I don't know if Peng et al 2022 or any other papers have studied this or not, but for this paper, it is necessary to explain how and why such a long-lived delaminated body can still be detected as high-velocity anomalies.

Second, did you consider the mantle convection ("mantle wind") that could push the delaminated body away? I noticed that in Peng et al 2022's 2D models, no subduction or mid-ocean ridge spreading is involved (refer to their model setup in Figure 1). In other words, there is little lateral motion ("mantle wind") in the convective mantle. But, in the real world, the asthenosphere under South China could be affected by the Philippine Sea plate subduction, Pacific subduction, India-Asia collision, opening of the South China Sea, and the Hainan plume/hotspot. All these processes could induce mantle flow or small-scale convection that drives the delaminated body away.

Third, did you consider the lateral drifting of the South China block (i.e., Eurasian plate)? In plate reconstruction models (such as the EarthByte group's model), the Eurasian plate has been constantly moving and rotating since the Triassic. In other words, South China at 180 Ma may not share the same position as today.

(3) The geological data are more or less disconnected from the geophysical result:

First, while Figure 4 is a nice compilation of the Sr-Nd isotope data of the Mesozoic mafic rocks, how this plot connects with the seismic tomography and thus supports the delamination model is still unclear. It seems that time is an important variable in the discussion (Line 207-216). So I am wondering if the authors can plot and interpret an age-eNd diagram, to see if the varying mantle sources can be better visualized.

Second, the authors use "Basin and Range-style magmatic province" to describe the magmatism in South China. So what exactly is the characteristic of the Basin and Range-style magmatism? Can you give a definition in the text and show its key features in the Sr-Nd and/or age-eNd plots? To be honest, I was surprised to find out that the authors did not cite any paper that studied the

magmatism in the Basin and Range Province of western North America.

Third, I would like to point out that in the western US, before the extension and magmatism in the Basin and Range Province, there was an orogenic plateau called "Nevadaplano" that occupied the same region (DeCelles 2004 *AJS* <https://doi.org/10.2475/ajs.304.2.105>). It is usually considered one of the many results of the Farallon flat-slab subduction (e.g., Liu et al 2008 *Science* <https://doi.org/10.1126/science.1162921>; Copeland et al 2017 *Geology* <https://doi.org/10.1130/G38810.1>). So, at least in a North American geologist's eyes, calling a magmatic province "Basin and Range-style" but opposing an earlier flat-slab subduction event is difficult to follow. Any comment on this dilemma?

3. A minor issue/suggestion on LV3 (Line 106): It is not wrong to describe the location of LV3 as the northern edge of the Taiwan arc-continent collision belt, but, LV3 is actually more proximal to the Okinawa Trough, an active back-arc basin due to the northward subduction of the Philippine Sea Plate. Low-velocity anomalies are expected, and have been imaged, beneath the Okinawa Trough. See, for example, Fukao and Obayashi 2013 *JGR* (<https://doi.org/10.1002/2013JB010466>), their Figure 4, profiles A-C.

4. One issue with writing is that the authors' observations are sometimes mixed with the interpretations. For example, Line 112, my understanding is that here you are describing the tomography results ("observations"). But "partly reworked lithosphere" is an interpretation, mostly based on geochemical data that is not the result of this study. In scientific writing, people usually prefer to separate observations from interpretations. Therefore, I suggest using more descriptive terms to describe the seismic characteristics of the Cathaysia block (such as the thinner thickness and fainter Vp perturbations). Similarly, another example is Line 174, "reworked". (You may have your own definition of "reworked lithosphere"; if so, please define it clearly in the manuscript.)

5. Suggestions on Figure 2:

(1) I suggest adding a line, or at least marking 28.5N on the edge of the map, to show the location of the profile in Fig. 2c.

(2) Also, the subducting plate is NOT the Pacific; it should be the Philippine Sea plate. Refer to Peter Bird 2003 *G-cube*, <https://doi.org/10.1029/2001GC000252>, Figure 3.

(3) Please state in the figure caption that the longitude of the profile extends from 100- to 132(?) degree East, this is different from the map's longitude range (105- to 123-degree East). It took me a few minutes to realize this difference.

(4) Figures 2a and 2b are labeled "80 km" and "160 km", respectively. But in figure caption, they have a range (40-120 km and 120-180 km, respectively). Strictly speaking, are they interchangeable? And why "160 km", instead of "150 km"?

6. On Figure 3e:

(1) What does the white contour line mean? Moho depth? Need to clarify in figure caption.

(2) The "Vs gradient variation across the Moho" is not clearly defined, making Figure 3e hard to understand (I've checked both the main text and supplement). I assume there is a prescribed direction to measure the "gradient", but I am still confused: is it just downward, or do I need to consider the lateral direction? If it is measured downward across the Moho, then what is the depth of the start and end point, respectively? Such as 5 km above and 5 km below the Moho? With all these questions, I am unable to understand or interpret how and why the Vs gradient varies between 0.2 and 1.0 (and the unit is missing, but I assume it's "per second").

(3) Moreover, I don't understand why the authors show the Vs gradient across Moho here, instead of a slice of Vs velocity or perturbations at 120 km depth. Such an image would greatly help the readers understand (and evaluate your interpretation of) the spatial distribution of the "delaminated lithosphere" and its relationship with the overlying lithosphere. Can some derivatives from the Moho serve this goal? I don't think so.

7. On Figure 5:

(1) Does this cartoon represent some time in the past (such as Jurassic?) or the present? Please clarify in figure caption.

(2) Based on figure caption, how can you make a cartoon that contains both "paleo-Pacific subduction" and "Pacific subduction"?

(3) Once again, if you are referring to the westward subducting plate at present, it should be the Philippine Sea plate, not Pacific.

(4) Is the delaminated body going up or going down? I am confused because you mentioned "regaining buoyancy and relamination" in the text (Line 243).

8. Other minor issues:

Line 23: I suggest removing "clearly" because it is subjective.

Line 26: I don't think "data mining" is appropriate here, based on what is presented in the Discussions section. According to Wikipedia, data mining is "the process of extracting and discovering patterns in large data sets involving methods at the intersection of machine learning, statistics, and database systems". Yes, it is a buzzword, but it is not what was presented in the manuscript.

Line 23: high-velocity, with a hyphen.

Line 26: by "the" foundering of

Line 39: because of "the" small scale

Line 47: "and" is characterized by

Line 53: "~70 km" in the parentheses is ambiguous and could confuse the readers. Does it represent the amount of extension (laterally), or the amount of thinning (vertically), or the average thickness of the lithosphere? Please clarify the physical meaning of this value in the parentheses.

Line 53: remove one of the two dots

Line 55-57: the usage of "cf." seems problematic here. "Cf." is an abbreviation for the Latin word confer, meaning "compare". It indicates that the cited literature supports a different claim than the one just made, and it is worthwhile to compare the two claims and assess the difference. I think "e.g." is more appropriate here. Or, just cite the literature without any Latin abbreviations.

Line 58: is "a" widely-"held" view

Line 61: "lacking" should be "lacks"

Line 62: "reflects" is a bit awkward; in my understanding, the lack of consensus "results from the fact", or "is caused by the fact", that the mantle structure beneath South China has not been well resolved.

Line 62: I think "mantle structure" is more appropriate than just the "lithosphere" here, because imaging the sublithospheric mantle is as important as imaging the lithospheric mantle to assess the competing models; moreover, one of the selling points of your work is the image of the delaminated body within the sublithospheric mantle.

Line 90: similarly, I think "mantle" is more accurate than "lithosphere" in this sentence.

Line 91: "the" should be removed, and "mantle" can also be removed if you adopt my previous suggestion.

Line 92: has "a" spatial resolution of

Line 97: "to exist" can be removed to keep the sentence concise

Line 100-101: please cite some relevant geological literature to support your interpretation ("Ar-Pt root & multiphase reworking")

Line 103: "develop" is not accurate, in my understanding. Try "coincide"?

Line 104-105: LV1 "is" located "beneath" the JSF ... LV2 "beneath" the TLF

Line 109: please remove "clearly" because it is subjective and unnecessary. When something is really clear, people can see it.

Line 120: for "the" south China lithosphere

Line 124: "Thanks" to "the" complementary sensitivities

Line 137: high-velocity, with a hyphen

Line 142: "becomes"

Line 142: change "from NE to SW" to "southwestward" to make it concise

Line 152: remove "the": Compared with profile B-B'

Line 152: is located in "the" deeper mantle

Line 154: "A" similar image has also been shown

Line 156: "keeps", not "kept"

Line 161: change this sentence to "See text for geological interpretations."

Line 173: again, please remove "clearly"

Line 179: Not "Pacific", it should be "Philippine Sea" plate.

Line 188: there are two ways to understand the word "gently": (1) the rate of the subduction is low (i.e., soft and gentle), (2) the dip of the subducting plate is low (i.e., low-angle or flat-slab

subduction). If a reader is not familiar with Li and Li 2007 Geology, he or she could feel confused. So, please rephrase it here.

Line 194: previously "been" interpreted as "the" far-distance effect

Line 216: Again, "Pacific" plate is problematic. The Cenozoic evolution of Southeast Asia (including the SE China coast) is much more complicated than the "Pacific retreat". Please refer to Wu et al 2016 JGR (<https://doi.org/10.1002/2016JB012923>)

Line 243: regain buoyancy, remove "the"

Line 256: yielded higher MgO and Cr, remove "the"

Line 276: "generated" should be "generating"

Line 278: "driving force"

Line 283: "on" the left side

Line 284: "the" one "on" the right side

RESPONSE TO REVIEWERS' COMMENTS

In this response, the original comments are marked in black regular font, our responses in blue regular font, and cited sentences of the revised version in *blue italic* font.

Reviewer #1 (Remarks to the Author):

Whether the partial removal of the deep lithospheric root in South China is caused by subduction or delamination remains a significant controversy. The authors, using a newly developed joint inversion algorithm of seismic body wave arrival times, surface wave dispersion curves, and teleseismic receiver functions, clearly imaged the structural features of the lithosphere-asthenosphere mantle in the lower part of the South China region. The study found that a series of the delaminated lithosphere beneath the convergent belt between the Yangtze and Cathaysia blocks provides very significant evidence for further study of the lithospheric thinning process in south China (even for other ancient Craton). The research will be of interest to the geological and geophysical community. However, there are still some problems that need to be addressed. I therefore suggest publication in NC after minor revision.

Reply: We are very grateful for your constructive and positive comments, which are very helpful for us to improve the interpretations.

Here are my comments.

Comment.1) Lines 26-27: The authors suggest that the lithospheric delamination of the South China Block resulted from the eclogitized mafic lower crust. However, An important condition for eclogitization in the lower crust is lithosphere thickening during orogenesis. The whole South China orogenic event (or lithosphere thickening event) can only be traced back to the Meso-Neo proterozoic, Devonian and Triassic collisional processes between the Cathaysia and Yangtze blocks, while the effects of regional flat-slab subduction are ignored(Li Z X and Li X H, 2007 in Geology; Lijun Liu et al., 2021 in ESR). Therefore, it raises the question of why the lower crust of the thickening lithosphere during the Meso-Neo proterozoic, Devonian and Triassic collisions suddenly experienced eclogitization until 180-160 Ma and further led to the lithospheric delamination. Please give the reasonable mechanism that triggered lithospheric delamination due to eclogitization of 180-160 Ma.

Reply: We have rethought interpretations following comments from you and the 2nd reviewer. Now we think flat slab model of Li and Li (2007) also played an important role in causing lithosphere delamination, which caused initial crustal thickening and mantle lithospheric deformation & destabilization. Then delamination of the mantle lithosphere and crustal root would be a natural consequence during slab retreat afterward. The related text in the Section “Mechanism and evolution of lithospheric delamination” is as follows:

“Prior to the Jurassic extension in south China, widespread lithospheric shortening took place in Neoproterozoic (the Jiangnan orogeny; Shu et al., 2021), Early-Paleozoic (Wuyi-Yunkai orogeny; Shu et al., 2021) and Triassic (intracontinental orogeny due to the far-field stress propagation between the Qinling-Dabie orogeny and Indosinian orogeny; Meng and Zhang, 1999; Song et al., 2017). This multiphase history of crustal thickening and lithospheric

shortening should have primed the development of 180-160 Ma mafic rocks and coeval A-type granitoids and potassic syenites in west Cathaysia (Zhou et al., 2006; Wang et al., 2008). The continental roots beneath both the Jiangnan and Dabie orogens partly remain at the present (Figs. 3b,e), supporting that orogenic collapse may have triggered the delamination.

Besides lithospheric thickening, the possible existence of an antecedent flat slab beneath south China could further facilitate the observed Jurassic tectonic activities. According to some previous studies, the flat slab foundered (Li and Li, 2007; Li et al., 2019) or rolled back (e.g. Zhou and Li, 2000; Dai et al., 2020) during the latest Triassic to earliest Jurassic. Numerical models showed that a flat slab could significantly deform and weaken the overriding lithosphere (Dai et al., 2020; Liu et al., 2021), thus creating a necessary condition for subsequent delamination (Elkins-Tanton, 2005; Hu et al., 2018; Peng et al., 2022). The progressive amount of lithospheric loss from the pan-Sichuan region to the southeast coast (Fig. 4) supports the role of this preceding flat slab that have caused more damage in the overriding lithosphere closer to the trench (Li and Li, 2007; Liu et al., 2021).”

Comment.2) Lines 56-58: The authors suggest that two existing mechanisms for lithospheric thinning in South China are contradictory, i.g. (1) the flat subduction and slab foundering and (2) the lithospheric delamination. However, the two views may not be contradictory in my opinion. They are two stages of one continuous process. Because slab foundering and lithospheric delamination may correspond to different stage of the lithospheric thinning processes. The migration pattern of magma, tectonic deformation, and metamorphic data together reveal that the initiation of flat subduction may have occurred between 260-250 Ma, while slab foundering occurred at 190-180 Ma, leading to widespread extension of the South China Block. This model matches well all present-day geological observations (including those mentioned in this manuscript) (Li and Li, 2007; Li S.Z., et al, 2019). This model has also been validated by geodynamic numerical simulations. Dai (2020) used dynamical numerical simulations to well validate the possibility of the slab foundering after the flat subduction. Meanwhile, Flat-slab subduction maybe provides two key kinetic conditions for the delamination of the South China lithosphere: (1) the flat-slab subduction process can lead to significant lithospheric thickening of the South China Block (Dai et al., 2020), providing temperature-pressure conditions for the eclogitization of the lower crust, which in turn triggers the delamination of the lithospheric mantle; (2) the flat-slab subduction process can lead to significant thinning and disruption of the lithospheric mantle of the overlying plate (Axen et al., 2018), which may provide favorable conditions for the continued delamination of the overlying slab lithosphere after flat-slab subduction and slab foundering.

Reference:

- [1] Li, Z.X., & Li, X.H. (2007). Formation of the 1300-km-wide intracontinental orogen and postorogenic magmatic province in mesozoic south china: a flat-slab subduction model. *Geology*, 35(2), 179-182.
- [2] S.Z. Li, X.Y Li, J. Zhou, H.H. Cao, et al.(2019). Mesozoic tectono-magmatic response in the East Asian ocean-continent connection zone to subduction of the Paleo-Pacific Plate. *Earth-Science Reviews* 192
- [3] Dai, L., Wang, L., Lou, D., Li, Z., & Yu, S., (2020). Slab rollback versus delamination: contrasting fates of flat-slab subduction and implications for south china evolution in the Mesozoic. *Journal of Geophysical Research- Solid Earth*, 125(4).
- [4] Axen, G.J., van Wijk, J.W. & Currie, C.A. (2018). Basal continental mantle lithosphere displaced by flat-slab subduction. *Nature Geosci* 11, 961–964.

Reply: Thanks so much for this comment. This really helps us to expand our interpretations on the mechanism and evolution of lithosphere delamination in south China. We have fully adopted the view that flat slab subduction also plays an important role on causing lithosphere delamination. Please see revised section “Mechanism and evolution of lithospheric delamination” and replies to the first comment. Specifically, the role of flat slab subduction is depicted in the revised version as follows:

“Besides lithospheric thickening, the possible existence of an antecedent flat slab beneath south China could further facilitate the observed Jurassic tectonic activities. According to some previous studies, the flat slab foundered (Li and Li, 2007; Li et al., 2019) or rolled back (e.g. Zhou and Li, 2000; Dai et al., 2020) during the latest Triassic to earliest Jurassic. Numerical models showed that a flat slab could significantly deform and weaken the overriding lithosphere (Dai et al., 2020; Liu et al., 2021), thus creating a necessary condition for subsequent delamination (Elkins-Tanton, 2005; Hu et al., 2018; Peng et al., 2022). The progressive amount of lithospheric loss from the pan-Sichuan region to the southeast coast (Fig. 4) supports the role of this preceding flat slab that have caused more damage in the overriding lithosphere closer to the trench (Li and Li, 2007; Liu et al., 2021).”

Comment.3) 3.1 Lines 240-244: The manuscript shows the critical tomographic evidence of lithospheric mantle delamination in South China, but it has not provided powerful evidence that the delamination must have started at 180-160 Ma.

Reply: In the revised version, we have clearly stated why the lithospheric delamination most likely initiated at 180-160 Ma based on the following facts:

- (1) South China experienced regional subsidence at 180-170 Ma and rapid uplift at ca. 160 Ma and such fast topographic changes and crustal deformation can be attributed to lithospheric foundering (Andersen et al., 2022).
- (2) The minor mafic rocks show two distinct mantle sources, and yielded a temporal source transition from “older” coexistence of depleted convective mantle and enriched lithospheric mantle at 180-160 Ma to “younger” dominance of enriched lithospheric mantle at 160-95 Ma and to the final coexistence of these two sources at 95-80 Ma (Fig. 5a,b).

3.2 On the contrary, in Fig. 3 B-B', does the delaminated lithosphere at 120 km depth mean that the South China lithospheric mantle is still undergoing delamination?”

Reply: Based on the further analysis of seismic images and geologic records, we believe the imaged detached lithosphere at 120 km depth represents the relaminated lithosphere after the Mesozoic delamination. The related text in the revised version is as follows:

“Although the present lithospheric structures, especially the cold and strong parts, could largely reflect the state after Mesozoic deformation, features that are clearly decoupled from the intact lithosphere, such as the detached lower lithosphere below south China below

(>95 km depth in Fig.4), should be subject to change over time. Our seismic images reveal that the top of the delaminated lithosphere lies approximately at ~90-120 km depth, which seems too shallow and unrealistic for delamination at 180-160 Ma. Recent geodynamic simulations suggest that the delaminated lithospheric segments could regain buoyancy and eventually relaminate to the base of the overlying lithosphere within 100-300 Myr (Peng et al., 2022). The observation that the geographic region of the Mesozoic magmatic province (Fig. 1b) is significantly broader than that of the presently imaged detached lower lithosphere (Fig. 4) implies that most of the delaminated lithosphere materials were lost into the deep mantle. The preserved portions of the lower lithosphere are either laterally connected with the thick Sichuan cratonic root on the NW or vertically with the upper lithosphere at some locations (Fig. 4). The mechanical coupling among these features should have kept the Mesozoic delaminated lower lithosphere from drifting away during subsequent relamination to their original locations (Peng et al., 2022).”

3.3 Besides, in Fig. 1b, there are two mafic rock data located near the JHP Fault (95-80 Ma) (in the manuscript, the authors do not explain the genesis of these two data), which happen to be located above the lithospheric delamination of the South China orogenic belt, does this mean that the beginning of delamination can be fixed at 95-80 Ma? Therefore, more evidence is still needed to verify the timing of this delamination event.

Reply: The two locations are from the Luosishan basalt in central Jiangxi Province (90.2 ± 0.3 Ma, Ar-Ar plateau age; Peng TP et al., 2004) and the Chunhuashan basalt in southern Jiangxi Province (83.3 ± 1.0 Ma, K-Ar age; Wang YJ et al., 2003), respectively. They both have OIB (ocean island basalt) -like geochemical features and are indicative for melting of upwelling asthenosphere. But their occurrences do not represent the beginning of delamination. In contrast, they, together with other older Jurassic and Early Cretaceous mafic rocks, may suggest the continuous delamination since 180-160 Ma. We have emphasized the timing of the delamination event in the revised text.

References:

Peng, T.P., Wang, Y.J., Jiang, Z.M., Yu, X.B., Peng, B.X., 2004. $^{40}\text{Ar}/^{39}\text{Ar}$ geochronology and geochemistry of Cretaceous basaltic rocks for the central and northwestern Jiangxi Province. 33(5), 447-458.

Wang, Y.J., Fan, W.M., Guo, F., Peng, T.P., Li, C.W., 2003. Geochemistry of Mesozoic mafic rocks adjacent to the Chenzhou-Linwu fault, South China: implications for the lithospheric boundary between the Yangtze and Cathaysia blocks. 45:3, 263-286, DOI: 10.2747/0020-6814.45.3.263.

Comment.4) In the C-C' of Fig. 3, the detached lithospheric mantle is completely decoupled from the original lithospheric mantle, while in the B-B' is a weak connection. I suggest the authors can provide a tomographic image of the middle position of these two sections. If the continuous detachment process can be identified, it can well support the authors' important conclusion about the extension of detachment from SW to NE.

Reply: Thanks for this suggestion. We have included one more profile between BB' and CC'. Indeed it shows the transition from weak connection to fully disconnection. We also included a new figure to show the 3D view of detached lithosphere deeper than 90 km.

Comment.5) In the manuscript, the authors suggest that the lithospheric mantle less than 70km depth has not been completely detached, which resulted from the ecologized mafic lower crust. However, if the ecologized mafic lower crust causes the delamination, the whole lithospheric mantle should be detached into the asthenosphere mantle, not only the lithospheric mantle deeper than 70 km. Therefore, it is controversial that the lithospheric delamination of South China resulted from the ecologized mafic lower crust.

Reply: We think this is because only the locally thickened crustal root would delaminate, a process that can entrain only part of the mantle lithosphere underneath, with other parts of the mantle lithosphere remain intact. The related text is as follows:

“It should be noted that only the locally thickened crustal root would delaminate, a process that can entrain only part of the mantle lithosphere underneath, with other parts of the mantle lithosphere remain intact.”

Comment.6) line 53: Wrong format,two periods repeat

Corrected.

Comment.7) Figure 1 should be improved, there are many detailed geological or tectonic maps of the South China Block and its neighbors, I prefer to recommend Sanzhong Li et al.'s paper on South China Block for you.

Thanks for the suggestion. The purpose of Figure 1 is to provide tectonic framework for this study. Therefore we did not include more detailed geological and tectonic details.

Reviewer #2 (Remarks to the Author):

NCOMMS-22-36882A-Z

In manuscript titled “Lithospheric delamination evidenced from seismic imaging and its controls on the Mesozoic Basin and Range Magmatic Province in south China”, Zhang and colleagues present a new seismic tomography of the lithosphere and asthenosphere under South China and report a group of newly identified high-velocity anomalies that are either dangling at the base of the thin lithosphere or floating in the asthenosphere. They further assess two competing hypotheses, flat-slab subduction vs. delamination, using the spatial distribution and geochemical signature of the Mesozoic igneous rocks in South China. Their favored interpretation is delamination, and they advocate that the magmatism and lithospheric modification in South China imitate the processes that took place in the Basin and Range Province of the western US.

The new tomography derived from the new joint inversion method of Han et al 2022 is a well-thought, rigorous, and state-of-the-art model. The authors have successfully discovered some fine details in the lithosphere and asthenosphere. I therefore believe that

this model is worthy of publication. However, the geological/tectonic interpretation is relatively weak, with many loose ends. My concerns are listed below. The first two are the major ones.

My recommendation is Reject, but encourage to resubmit. I would be happy to review this manuscript again.

Reply: Thanks very much for your constructive comments, which are very helpful for us to improve our geological/tectonic interpretation based on new velocity model and geochemical data.

1. On the evaluation of the flat-slab subduction model:

In Line 188-192, the authors wrote that (1) Li and Li 2007 proposed that flat-slab subduction took place before 170 Ma, and (2) the igneous rocks emplaced between 180-170 Ma suggest an extensional environment. Based on the temporal relationship, the authors ruled out the flat-slab subduction model.

Well, the statement of ">170 Ma" (Line 189) is not wrong, but I am afraid this is rather misleading. In their Figure 4, Li and Li (2007) explicitly showed that the flat-slab subduction peaked at 210 Ma (or around 200 Ma if you use the age-distance plot in their Figure 3). In their model, the flat slab began to break into pieces and fold "like a taco" (as per Eugene Humphreys in his Farallon flat-slab papers and talks) and sink into the convective mantle by 190 Ma. Please note that ">170 Ma" and "210-200 Ma" are two different concepts in this context. So, just like the Oligocene (~30 Ma) volcanism in the western US cannot be used to rule out the Farallon flat-slab subduction that peaked at ~50 Ma (see, for example, Copeland et al 2017 *Geology* <https://doi.org/10.1130/G38810.1>), the 180-170 Ma igneous rocks in South China cannot be used to rule out the "Paleo-Pacific" flat-slab subduction that peaked at ~210-200 Ma.

I want to clarify that I am not forcing the authors to support the flat-slab subduction model. Instead, my goal here is to help the authors develop an unbiased evaluation of a model. Otherwise, if this is published, anybody could easily write a comment to attack this flaw.

Reply: We really appreciate your comment on flat-slab subduction model, which has helped us to rethink our original interpretation. In the revised text, "We suggest that the Mesozoic tectonism in south China may reflect the composite effect of multiple geodynamic processes including abnormal subduction and lithospheric delamination, and that the Jurassic crustal extension and magmatism mainly resulted from the latter." The relationship between flat slab subduction and lithosphere delamination has been described as follows:

"Besides lithospheric thickening, the possible existence of an antecedent flat slab beneath south China could further facilitate the observed Jurassic tectonic activities. According to some previous studies, the flat slab foundered (Li and Li, 2007; Li et al., 2019) or rolled back (e.g. Zhou and Li, 2000; Dai et al., 2020) during the latest Triassic to earliest Jurassic. Numerical models showed that a flat slab could significantly deform and weaken the overriding lithosphere (Dai et al., 2020; Liu et al., 2021), thus creating a necessary condition for subsequent delamination (Elkins-Tanton, 2005; Hu et al., 2018; Peng et al., 2022). The progressive amount of lithospheric loss from the pan-Sichuan region to the southeast coast (Fig. 4) supports the role of this preceding flat slab that have caused more damage in the overriding lithosphere closer to the trench (Li and Li, 2007; Liu et al., 2021)."

2. On the evaluation of the delamination model:

Let's put the flat-slab subduction model aside and say that the delamination model is correct. But three aspects still concern me.

(1) What caused the delamination, and is it plausible?

The authors suggested that the delamination started at 180-160 Ma and expanded from SW to NE, due to orogenic (or "post-orogenic"?) collapse. Since the most recent orogeny by 180 Ma was the Triassic Qinling-Dabie in the north, why did the delamination begin in the south, more than 500 km away from the Qinling-Dabie orogenic belt? Did the delaminations under Sierra Nevada, Colorado Plateau, or Canadian Rocky Mountains share a similar feature? Are there any numerical modeling studies that suggest such a "far-field effect of delamination"? I am not aware of such things. If there is any, please point it out in the manuscript.

Reply: There exist actually two orogens in south China, the Jiangnan orogen in south and the Qinling-Dabie orogen in northeast. The reason that delamination started at 180-160 Ma and expanded from SW to NE was mainly controlled by Jiangnan orogen during early Jurassic.

The related text is as follows:

"The joint analysis of our high-resolution tomography images (Figs. 2, 3, 4) and geological observations (Figs. 1b, 5) outlines a consistent picture (Fig. 6) of Mesozoic lithospheric delamination and resulting decompression melting below south China: the process started below Jiangnan orogen during early Jurassic, and subsequently expanded outwards, with the final stage of delamination terminated in the NE during early Cretaceous. The post-120 Ma magmatism could have resulted from asthenosphere upwelling due to rollback of the paleo-Pacific slab. The peel-off style of the lower-lithosphere delamination and associated upwelling hot asthenosphere (Fig. 6a) exert significant extension on the overlying lithosphere, leading to the Basin and Range-style crustal deformation and magmatism. This delamination style is further reflected in the present-day lithosphere configuration (Figs. 3, 4, 6b). We argue that the initial delamination at 180-160 Ma along the convergent boundary between the Yangtze and Cathaysia blocks produced a limited scale of lower-lithosphere drips, where both the large depth and the weak asthenospheric upwelling prohibited extensive melting (Drew et al., 2009), thus generating small-volume mafic magmas. Subsequently thinned and fractured lithosphere allow more efficient adiabatic upwelling that provided enough driving-force for lithospheric extension and extensive melting of the overlying crust, forging a NE-trending Basin-Range-style Late-Mesozoic magmatic province."

(2) The complexities and uncertainties in delamination and relamination:

The tomography model shows the present-day mantle structure, and the authors focus on the Mesozoic tectonics/magmatism in the discussion. The authors also admitted (Line 240-244) that the delaminated body is too shallow and too long-lived in their favored model. Then they cited Peng et al 2022 EPSL to argue that this delaminated body has (or could have) regained buoyancy and could relaminate to the overlying lithosphere. I have three sub-questions regarding these arguments:

First, if a delaminated body has regained buoyancy and been heated by the ambient asthenospheric mantle for >100 Myr, will it still show as a high-velocity anomaly? I don't know if Peng et al 2022 or any other papers have studied this or not, but for this paper, it is necessary to explain how and why such a long-lived delaminated body can still be detected

as high-velocity anomalies.

Reply: Thanks very much for your thoughtful comments. For the relaminated lithosphere, once it is cooled down and it should still be shown as high velocity anomalies. We think this is the present state of these partially detached lithospheric portions.

Second, did you consider the mantle convection (“mantle wind”) that could push the delaminated body away? I noticed that in Peng et al 2022’s 2D models, no subduction or mid-ocean ridge spreading is involved (refer to their model setup in Figure 1). In other words, there is little lateral motion (“mantle wind”) in the convective mantle. But, in the real world, the asthenosphere under South China could be affected by the Philippine Sea plate subduction, Pacific subduction, India-Asia collision, opening of the South China Sea, and the Hainan plume/hotspot. All these processes could induce mantle flow or small-scale convection that drives the delaminated body away.

Third, did you consider the lateral drifting of the South China block (i.e., Eurasian plate)? In plate reconstruction models (such as the EarthByte group’s model), the Eurasian plate has been constantly moving and rotating since the Triassic. In other words, South China at 180 Ma may not share the same position as today.

Reply: After further examining the seismic images and geologic/geochemical data, we found that the detached lower lithosphere is actually laterally connected with the thick Sichuan cratonic root on the NW or vertically with the upper lithosphere at some locations.

Therefore, the mechanical coupling among these features should have kept the Mesozoic delaminated lower lithosphere from drifting away during subsequent relamination to their original locations (Peng et al., 2022). Therefore, our interpretation is largely independent of mantle flow and South China movement since the Mesozoic. That said, we did acknowledge the fact that some fully delaminated materials would not have relaminated to their original locations, where lateral offset would occur (as shown in the updated Figure 5).

The related revised text is as follows:

“Recent geodynamic simulations suggest that the delaminated lithospheric segments could regain buoyancy and eventually relaminate to the base of the overlying lithosphere within 100-300 Myr (Peng et al., 2022). The observation that the geographic region of the Mesozoic magmatic province (Fig. 1b) is significantly broader than that of the presently imaged detached lower lithosphere (Fig. 4) implies that most of the delaminated lithosphere materials were lost into the deep mantle. The preserved portions of the lower lithosphere are either laterally connected with the thick Sichuan cratonic root on the NW or vertically with the upper lithosphere at some locations (Fig. 4). The mechanical coupling among these features should have kept the Mesozoic delaminated lower lithosphere from drifting away during subsequent relamination to their original locations (Peng et al., 2022).”

(3) The geological data are more or less disconnected from the geophysical result: First, while Figure 4 is a nice compilation of the Sr-Nd isotope data of the Mesozoic mafic rocks, how this plot connects with the seismic tomography and thus supports the delamination model is still unclear. It seems that time is an important variable in the discussion (Line 207-216). So I am wondering if the authors can plot and interpret an age-

ϵ_{Nd} diagram, to see if the varying mantle sources can be better visualized.

Reply: Following your suggestion, we have revised original Figure 4 (now Figure 5) to include $\epsilon_{Nd}(t)$ -age variations of for the rock samples, as follows:

Second, the authors use “Basin and Range-style magmatic province” to describe the magmatism in South China. So what exactly is the characteristic of the Basin and Range-style magmatism? Can you give a definition in the text and show its key features in the Sr-Nd and/or age- ϵ_{Nd} plots? To be honest, I was surprised to find out that the authors did not cite any paper that studied the magmatism in the Basin and Range Province of western North America.

Reply: This name is borrowed from Li and Li (2007) and we have included a reference for the magmatism in the Basin and Range Province in western United States, as follows:

“The South China block in east Asia formed by the Neoproterozoic amalgamation of the Yangtze and Cathaysia blocks (Fig. 1), and is characterized by the development of the world-class Mesozoic Basin and Range-style magmatic province (Li and Li, 2007), which physiographically is similar to the Basin and Range Province in the western United States where synextensional magmatism was developed (Gans, 1989).”

An earlier pioneer on South China basin and range tectonics is Gilder who addressed the term in their Tectonophysics paper (Gilder et al., 1991). Typical basin and range-style magmatic province is characterized by extensional basin and range tectonics, and such geological features are also very common in the southeast China (Shu LS et al., 2006). The basin and range tectonics in western United States took place later than that to the western margin of Paleo-Pacific Ocean Plate. In addition, Sr and Nd isotopes plot mainly reflects the magmatism source properties, rather than tectonic settings. So, we do not think it is necessary to plot and compare the Sr and Nd isotopes of the two areas and use the plot to explain the magmatic characters of basin and range-style magmatic provinces.

Gilder, S.A., Keller, G.R., Luo, M., and Goodell, P.C., 1991, Timing and spatial distribution of rifting in China: Tectonophysics, v. 197, p. 225–243, doi: 10.1016/0040–1951(91)90043-R

Third, I would like to point out that in the western US, before the extension and magmatism in the Basin and Range Province, there was an orogenic plateau called “Nevadaplano” that occupied the same region (DeCelles 2004 *AJS* <https://doi.org/10.2475/ajs.304.2.105>). It is usually considered one of the many results of the Farallon flat-slab subduction (e.g., Liu et al 2008 *Science* <https://doi.org/10.1126/science.1162921>; Copeland et al 2017 *Geology* <https://doi.org/10.1130/G38810.1>). So, at least in a North American geologist’s eyes, calling a magmatic province “Basin and Range-style” but opposing an earlier flat-slab subduction event is difficult to follow. Any comment on this dilemma?

Reply: Thanks for pointing this out. Now in the revised version, we included flat-slab subduction model as an earlier stage for lithosphere delamination.

3. A minor issue/suggestion on LV3 (Line 106): It is not wrong to describe the location of LV3 as the northern edge of the Taiwan arc-continent collision belt, but, LV3 is actually more proximal to the Okinawa Trough, an active back-arc basin due to the northward subduction of the Philippine Sea Plate. Low-velocity anomalies are expected, and have been imaged, beneath the Okinawa Trough. See, for example, Fukao and Obayashi 2013 *JGR* (<https://doi.org/10.1002/2013JB010466>), their Figure 4, profiles A-C.

Reply: Thanks for pointing this out. We have included Fukao and Obayashi 2013 *JGR* paper in our reference list.

4. One issue with writing is that the authors' observations are sometimes mixed with the interpretations. For example, Line 112, my understanding is that here you are describing the tomography results ("observations"). But "partly reworked lithosphere" is an interpretation, mostly based on geochemical data that is not the result of this study. In scientific writing, people usually prefer to separate observations from interpretations. Therefore, I suggest using more descriptive terms to describe the seismic characteristics of the Cathaysia block (such as the thinner thickness and fainter Vp perturbations). Similarly, another example is Line 174, "reworked". (You may have your own definition of "reworked lithosphere"; if so, please define it clearly in the manuscript.)

Reply: Thanks for pointing this out. We have corrected these cases to avoid observations with interpretations.

5. Suggestions on Figure 2:

(1) I suggest adding a line, or at least marking 28.5N on the edge of the map, to show the location of the profile in Fig. 2c.

Reply: We have marked 28.5N on the edge of the map.

(2) Also, the subducting plate is NOT the Pacific; it should be the Philippine Sea plate. Refer to Peter Bird 2003 G-cube, <https://doi.org/10.1029/2001GC000252>, Figure 3.

It is changed.

(3) Please state in the figure caption that the longitude of the profile extends from 100- to 132(?) -degree East, this is different from the map's longitude range (105- to 123-degree East). It took me a few minutes to realize this difference.

Added.

(4) Figures 2a and 2b are labeled "80 km" and "160 km", respectively. But in figure caption, they have a range (40-120 km and 120-180 km, respectively). Strictly speaking, are they interchangeable? And why "160 km", instead of "150 km"?

In the revised caption, it is now stated as 80 km and 160 km. The original purpose is to state that the velocity depth slices mainly reflect the velocity feature within two depth ranges.

6. On Figure 3e:

(1) What does the white contour line mean? Moho depth? Need to clarify in figure caption.

Reply: The solid black line denotes the Moho.

(2) The "Vs gradient variation across the Moho" is not clearly defined, making Figure 3e hard to understand (I've checked both the main text and supplement). I assume there is a prescribed direction to measure the "gradient", but I am still confused: is it just downward, or do I need to consider the lateral direction? If it is measured downward across the Moho, then what is the depth of the start and end point, respectively? Such as 5 km above and 5 km below the Moho? With all these questions, I am unable to understand or interpret how and why the Vs gradient varies between 0.2 and 1.0 (and the unit is missing, but I assume it's "per second").

Reply: In revised text, we explained how Vs gradients across the Moho are calculated, "From the Vs model, Vs gradients across the Moho can be derived from Vs values at nodes immediately above and below the actual Moho interface, which have the unit of 1/s."

Yes, the unit for the Vs gradient is 1/s and has been added to the colorbar.

(3) Moreover, I don't understand why the authors show the Vs gradient across Moho here, instead of a slice of Vs velocity or perturbations at 120 km depth. Such an image would greatly help the readers understand (and evaluate your interpretation of) the spatial distribution of the "delaminated lithosphere" and its relationship with the overlying lithosphere. Can some derivatives from the Moho serve this goal? I don't think so.

Reply: Thanks for this suggestion. We have included a 3D plot showing delaminated lithosphere below 90 km in new Figure 4 and also depth slices of 90, 110, 120 and 130 km in supplementary materials. These new plots really helped us better understand how delaminated/relaminated lithosphere blobs are spatially distributed.

7. On Figure 5:

(1) Does this cartoon represent some time in the past (such as Jurassic?) or the present? Please clarify in figure caption.

(2) Based on figure caption, how can you make a cartoon that contains both "paleo-Pacific subduction" and "Pacific subduction"?

(3) Once again, if you are referring to the westward subducting plate at present, it should be the Philippine Sea plate, not Pacific.

(4) Is the delaminated body going up or going down? I am confused because you mentioned "regaining buoyancy and relamination" in the text (Line 243).

Reply: Following your comments, this figure has been remade to show scenarios at Mesozoic and present, respectively.

Figure 6. Illustration of south China tectonic evolution since the Jurassic. (a) Late Mesozoic (180-70 Ma) lithosphere delamination, asthenosphere upwelling and paleo-Pacific subduction in south China. The asthenosphere upwelling and decompression melting during 180-120 Ma were triggered by the lithospheric delamination, whereas those after 120 Ma were caused by rollback of the paleo-Pacific subducting slab. (b) The restored lithospheric structure after relamination as imaged today.

8. Other minor issues:

Line 23: I suggest removing "clearly" because it is subjective.

Removed.

Line 26: I don't think "data mining" is appropriate here, based on what is presented in the Discussions section. According to Wikipedia, data mining is "the process of extracting and discovering patterns in large data sets involving methods at the intersection of machine learning, statistics, and database systems". Yes, it is a buzzword, but it is not what was presented in the manuscript.
Mining is now removed.

Line 23: high-velocity, with a hyphen.
Added.

Line 26: by "the" foundering of
Added.

Line 39: because of "the" small scale
Added.

Line 47: "and" is characterized by
Added.

Line 53: "~70 km" in the parentheses is ambiguous and could confuse the readers. Does it represent the amount of extension (laterally), or the amount of thinning (vertically), or the average thickness of the lithosphere? Please clarify the physical meaning of this value in the parentheses.
Changed to "to present depths of ~70 km"

Line 53: remove one of the two dots
Removed

Line 55-57: the usage of "cf." seems problematic here. "Cf." is an abbreviation for the Latin word confer, meaning "compare". It indicates that the cited literature supports a different claim than the one just made, and it is worthwhile to compare the two claims and assess the difference. I think "e.g." is more appropriate here. Or, just cite the literature without any Latin abbreviations.
Changed to "e.g."

Line 58: is "a" widely-"held" view
Changed

Line 61: "lacking" should be "lacks"
Changed

Line 62: "reflects" is a bit awkward; in my understanding, the lack of consensus "results from the fact", or "is caused by the fact", that the mantle structure beneath South China has not been well resolved.
This sentence has been changed to the following:
"The existing debates are at least partly due to the poorly resolved upper mantle structure of south China (Gao et al., 2022 and references therein) that hampers understanding the relationship between deep geodynamic processes and surface magmatism remains."

Line 62: I think "mantle structure" is more appropriate than just the "lithosphere" here, because imaging the sublithospheric mantle is as important as imaging the lithospheric mantle to assess the competing models; moreover, one of the selling points of your work is the image of the delaminated body within the sublithospheric mantle.

Changed to "upper mantle structure"

Line 90: similarly, I think "mantle" is more accurate than "lithosphere" in this sentence.

Changed

Line 91: "the" should be removed, and "mantle" can also be removed if you adopt my previous suggestion.

This sentence is now changed as follows:

"To generally understand the velocity structure of the mantle in the study region, we have conducted large-scale mantle seismic tomography in south China"

Line 92: has "a" spatial resolution of

Changed

Line 97: "to exist" can be removed to keep the sentence concise

Removed

Line 100-101: please cite some relevant geological literature to support your interpretation ("Ar-Pt root & multiphase reworking")

Li et al., (2021) is added

Line 103: "develop" is not accurate, in my understanding. Try "coincide"?

Changed to "They roughly occur..."

Line 104-105: LV1 "is" located "beneath" the JSF ... LV2 "beneath" the TLF

Changed

Line 109: please remove "clearly" because it is subjective and unnecessary. When something is really clear, people can see it.

Removed

Line 120: for "the" south China lithosphere

Added

Line 124: "Thanks" to "the" complementary sensitivities

Changed

Line 137: high-velocity, with a hyphen

Added

Line 142: "becomes"

Corrected

Line 142: change "from NE to SW" to "southwestward" to make it concise
Changed

Line 152: remove "the": Compared with profile B-B'
Changed

Line 152: is located in "the" deeper mantle
Changed

Line 154: "A" similar image has also been shown
Changed

Line 156: "keeps", not "kept"
Changed

Line 161: change this sentence to "See text for geological interpretations."
Changed to "See text for more interpretations"

Line 173: again, please remove "clearly"
Removed

Line 179: Not "Pacific", it should be "Philippine Sea" plate.
Changed

Line 188: there are two ways to understand the word "gently": (1) the rate of the subduction is low (i.e., soft and gentle), (2) the dip of the subducting plate is low (i.e., low-angle or flat-slab subduction). If a reader is not familiar with Li and Li 2007 Geology, he or she could feel confused. So, please rephrase it here.
This sentence has been rephrased as "...the paleo-Pacific plate subducted flatly beneath the south China..."

Line 194: previously "been" interpreted as "the" far-distance effect
Changed

Line 216: Again, "Pacific" plate is problematic. The Cenozoic evolution of Southeast Asia (including the SE China coast) is much more complicated than the "Pacific retreat". Please refer to Wu et al 2016 JGR (<https://doi.org/10.1002/2016JB012923>)
This sentence is now rephrased as "...may indicate the effect of paleo-Pacific subduction at the latest Mesozoic (Zhou et al., 2006; Liu et al., 2021)."

Line 243: regain buoyancy, remove "the"
Changed

Line 256: yielded higher MgO and Cr, remove "the"
Changed

Line 276: "generated" should be "generating"
Changed

Line 278: “driving force”
Changed

Line 283: “on” the left side
This sentence has been rewritten

Line 284: “the” one “on” the right side
This sentence has been rewritten

Thanks again for these detailed corrections that are very helpful for improving the readability of the revised version.

REVIEWER COMMENTS

Reviewer #1 (Remarks to the Author):

The revised draft has answered all my questions, especially, the new data (Fig.3 and Fig.4) clearly shows the deep structural characteristics of the South China block, which is perfectly combined with the geological evidence. Therefore, I suggest to accept this paper.

Reviewer #2 (Remarks to the Author):

NCOMMS-22-36882B

Zhang and co-authors have made substantial changes to their manuscript titled "Lithospheric delamination evidenced from seismic imaging and its controls on the Mesozoic Basin and Range Magmatic Province in south China". While I still believe that the geophysical results are worthy of publication, the reasoning and interpretation in this manuscript are still weak and disconnected from the seismic observations. Below I list four major concerns, followed by other minor issues and suggestions. My recommendation is Major Revision.

Major concerns:

1. In my understanding, the authors aimed to build a causal relationship between the seismic anomalies and the magmatic record. The seismic anomalies are at the local scale (on the order of 100s km). However, the geochemical data are compiled and plotted for entire South China, at the 1000-km scale. If delamination mainly occurred under the Nanling Range (south-central part of South China), then plotting the geochemical data from other areas such as Fujian and Zhejiang (easternmost South China) does not make sense to me. I suggest the authors plot only the igneous rocks above the delaminated/relaminated body to see if the geochemical data can be incorporated into the story. This is particularly important if the goal here is to tease out the effect of "Paleo-Pacific" oceanic subduction from the east.

2. Another major concern is objectiveness. I don't see a fair evaluation of the flat-slab subduction model by Li and Li 2007. Actually, I don't even know if the authors support or refute the flat-slab subduction model because the writing (such as Line 275-284) is not very clear to me. In another place in the manuscript, the authors wrote "However, the plausible geodynamic relationship among these early-Mesozoic tectonic processes remains largely unexplored" (Line 223-224). The authors offered no explanation, data, or reference to support this critique of the Li and Li 2007 model. I actually don't know how, from this sentence, the flat-slab model can be ruled out. Even from a philosophical point of view, the fact that something has not been explored cannot be used to negate its existence. The absence of evidence is not the evidence of absence. Moreover, for a research article (instead of a review), a common practice in science is to use the new observations to assess previous models (then the authors either modify the existing model(s) or abandon them and propose a new one). Therefore, as a reader, my expectation for this paper is to see a comparison of the newly identified high- and low-velocity anomalies against the predictions from the previous models. For a seismological paper, it is not a good idea to heavily (or even purely) rely on geochemical data to support or refute a model. Note that it is okay to admit that the new observations cannot fully rule out a model. The authors can still lean towards their preferred model, as long as the evaluation is unbiased. People understand this. However, a paper may lose credibility if the argument shows bias.

3. On the presentation and interpretation of the results. The authors now show more profiles and depth slices (Figs. 2, 3, S9, S10), allowing me to raise several questions.

(1) In Fig. 2, Vp model, LV1 is centered on 27N, 114E (Hunan-Jiangxi border) at 80 km and 160 km. In Fig. S9, the Vs velocity here is relatively low, but not as low as in many other areas. Why does this discrepancy occur? What does this imply?

(2) Figure S9 shows depth slices at 90-130 km. How do these depth slices support the statement

about the crust (~30 km) in Line 289-292? I don't get it.

(3) Line 158: "Between 114°E and 120°E, a small and faint high-velocity blob exists within the asthenosphere shown by low-velocity anomalies (LV2)". This sentence is rather confusing. Is the "high-velocity" blob shown by a "low-velocity" anomaly?

Also, I could only see a "small and faint high-velocity blob" at ~120 km depth at 116°E on profile B-B'. Using "Between 114°E and 120°E" to describe its location is not precise at all.

Also, if it is important and worthy of mention, please give a name to this small and faint high-velocity blob.

In addition, if this blob is worthy of mention, then how to prove that it is not an artifact?

(4) From Line 143, now the authors explain that the Vs gradient is measured in the vertical direction. Then I suggest the authors say more about the geological implications of Vs gradient. In other words, why is it important to calculate Vs gradient? What geological processes are associated with high Vs gradient, and what leads to low Vs gradient? For example, how does magma underplating at the Moho depth (such as those shown in Figure 6) affect the Vs gradient?

(5) Let's assume that Vs gradient is an indicator for delamination. In Figure 3F, the high Vs gradient zone trends NE-SW, roughly following the Jiangnan orogenic belt. However, in Figure 4, the interpreted delaminated/relaminated lower lithosphere (red isosurface along the east edge of the Yangtze cratonic lithosphere) is trending north-south. These two orientations differ by 60-70 degrees. This is a significant difference. Why?

(6) I suggest the authors make a movie to show the three-dimensional geometry of the high Vs anomalies. It is always challenging to use a piece of paper to show a 3D object. Therefore, a movie always helps. Showing more figures from different angles (in map view, cross-section view, and oblique view) in the supplementary file should also be useful to deliver the message to the readers.

4. Some problematic (or wrong) statements:

(1) The authors argue that "it is difficult for oceanic subduction that continued from Triassic to Cretaceous to explain" two major observations (Line 324-331). I think Prof. Zheng-Xiang Li's serial works over the past 2 decades had explained both points listed here, at least in the first order. See for example Li and Li 2007 *Geology* (please, read this paper again, so that you will see why the statement "magmatism *STARTED* in west Cathaysia at 180 Ma" (Line 326-327) is not faithful to the well-known geological record in South China).

See also:

Zheng-Xiang Li et al 2012 *Tectonophysics* <https://doi.org/10.1016/j.tecto.2012.02.011>;

Meng et al 2012 *Lithos* <https://doi.org/10.1016/j.lithos.2011.11.022>;

Xian-Hua Li et al 2012 *CG* <https://doi.org/10.1016/j.chemgeo.2011.10.027>;

Zhu et al 2017 *Lithos* <https://doi.org/10.1016/j.lithos.2017.10.008>;

Cui et al 2021 *GR* <https://doi.org/10.1016/j.gr.2021.06.021>.

(2) Line 52: It's unclear and unfair to say that "it remains unclear that ..." without detailed explanations. On what issues exactly does the relationship between delamination and other tectonics processes remain unclear? If this is not elaborated on, I can almost foresee that other readers who also work on this topic would feel offended. I don't think it's your intention here to belittle the previous works. So, I suggest rewriting this sentence.

(3) Line 117: The Taiwan island cannot be simply equivalent to "an orogenic belt". Instead, the island contains many more tectonic units and has experienced complex geologic history (such as the eastward subduction of the Eurasian plate under the Philippine Sea plate under most of the island, and the Luzon arc-continent collision). I am afraid that the authors are not familiar with Taiwan geology. So I suggest, again, that the authors take my previous suggestion and associate LV3 with the Okinawa backarc extension. (The East China Sea rifting is also part of the story here, but I think it can be omitted for now, just to keep things simple.) I could see here that the authors try to associate every low-velocity anomaly with a major fault or a plate boundary. If my speculation is correct, then associating LV3 with Okinawa makes perfect sense because the

present-day Eurasia-Philippine Sea plate boundary exists there.

(4) Line 154: The logic relation is problematic here. By using "due to", are you suggesting that the development of LV2 caused the lithosphere near Tanlu to break? Please keep in mind that correlation does not imply causation.

Also, this entire paragraph describes the observations. I suggest not including opinions or interpretations here. "Due to" means an opinion.

(5) In my concern 2(2) in the previous round of review, I asked if a "delaminated" body can still be detected after staying in the convective mantle and being warmed for 100+ Myr. The authors pointed out that a "relaminated" lithosphere, once cooled down, can be detected as a high-velocity anomaly. Their statement is correct on the "relaminated" lithosphere, but I cannot see how this helps to explain my question on the *delaminated* lithosphere.

5. Other minor issues and suggestions:

Line 56: Reference is needed here to support the Neoproterozoic Yangtze-Cathaysia amalgamation, because Li and Li 2007 Geology does not focus on this topic.

Line 57: Please start a new sentence at "which". This sentence is too long now. Long sentences tend to lower readability.

Line 77: remove "remains".

Line 96-97: please remove "extension" and add references (such as Li and Li 2007 Geology) to support this statement.

Figure 2(C): please use white color for the label "Yangtze". Also, explain what the white dots denote in the figure caption.

The paragraph that begins at Line 147: This long paragraph describes the seismological observations. However, it is not written concisely. The authors described every profile individually, making it too long to grasp the core observations. It would be okay to write in this way if this manuscript is submitted to JGR. But for a short paper like NC, a better practice is to point out a key anomaly in one and only one sentence and refer to the relevant panel(s) in Figure 3 (and supplementary figures). This way, the readers don't need to memorize all the details on every profile first to build an imaginary 3D model in their minds.

Line 245: "progressive" is more accurate than "steady" in this sentence. "A steady increase" may mean an increase at a steady rate over time, which may not be the case here.

In the title and throughout the manuscript, I suggest using "South China" (with an uppercase "S") instead of "south China". "South China" in this study refers to the specific continental block. When one wants to refer to the approximate geographic region, "southern China" is more commonly used (which is also more grammatically correct). Please refer to these two Wikipedia pages to get a general idea of the usage of South versus southern.

https://en.wikipedia.org/wiki/Northern_and_southern_China

https://en.wikipedia.org/wiki/South_China_Craton

In the Supplementary file, please check the spelling and grammar again. Some space is missing, such as "Qaidambasin" in Figure S1 caption.

In Figure S1, the "Philippine" Sea plate has a double "p" and its subduction polarity is northwestward. Please correct these two errors.

In Figure S3, please also show the input model for the resolution test. Also, why is there a low-velocity anomaly (around 28N 113E) at the center of the output model at 40 km?

Figure S10 mentions that adakitic rocks are shown by small solid dots. But I could only see cities shown by this symbol. The adakitic rocks are not plotted.

RESPONSE TO REVIEWERS' COMMENTS

Reviewer #1 (Remarks to the Author):

The revised draft has answered all my questions, especially, the new data(Fig.3 and Fig.4) clearly shows the deep structural characteristics of the South China block, which is perfectly combined with the geological evidence. Therefore, I suggest to accept this paper.

Response: Thank you very much for you very positive comments on our manuscript and acceptance of our revision.

Reviewer #2 (Remarks to the Author):

NCOMMS-22-36882B

Zhang and co-authors have made substantial changes to their manuscript titled “Lithospheric delamination evidenced from seismic imaging and its controls on the Mesozoic Basin and Range Magmatic Province in south China”. While I still believe that the geophysical results are worthy of publication, the reasoning and interpretation in this manuscript are still weak and disconnected from the seismic observations. Below I list four major concerns, followed by other minor issues and suggestions. My recommendation is Major Revision.

Response: Thank you very much for you overall positive comments. We carefully thought over your constructive comments and revised the manuscript accordingly. Your suggestions have been incorporated into the revised manuscript. The geochemical plot has been revised to reflect mainly the data from the west Cathaysia and parts (i.e. the middle and southwestern segments that are very close to the west Cathaysia) of the Jiangnan orogen. We also revised the data interpretation to avoid further confusion.

Major concerns:

1. In my understanding, the authors aimed to build a causal relationship between the seismic anomalies and the magmatic record. The seismic anomalies are at the local scale (on the order of 100s km). However, the geochemical data are compiled and plotted for entire South China, at the 1000-km scale. If delamination mainly occurred under the Nanling Range (south-central part of South China), then plotting the geochemical data from other areas such as Fujian and Zhejiang (easternmost South China) does not make sense to me. I suggest the authors plot only the igneous rocks above the delaminated/re-laminated body to see if the geochemical data can be incorporated into the story. This is particularly important if the goal here is to tease out the effect of “Paleo-Pacific” oceanic subduction from the east.

Response: Thanks for your valuable comments. In fact, our data are mainly from the south-central part of South China. Some samples from the coastal area (Zhejiang and Fujian provinces) included in the original plot do not change the overall trend shown in our previous figure.

According to your suggestion, we have removed the samples from the coastal area from the revised diagrams. The revised figure 5 shows more evident trends with time and more strongly support our statements in the main text. We have updated the relevant sentences accordingly.

2. Another major concern is objectiveness. I don't see a fair evaluation of the flat-slab subduction model by Li and Li 2007. Actually, I don't even know if the authors support or refute the flat-slab subduction model because the writing (such as Line 275-284) is not very clear to me.

In another place in the manuscript, the authors wrote "However, the plausible geodynamic relationship among these early-Mesozoic tectonic processes remains largely unexplored" (Line 223-224). The authors offered no explanation, data, or reference to support this critique of the Li and Li 2007 model. I actually don't know how, from this sentence, the flat-slab model can be ruled out. Even from a philosophical point of view, the fact that something has not been explored cannot be used to negate its existence. The absence of evidence is not the evidence of absence. Moreover, for a research article (instead of a review), a common practice in science is to use the new observations to assess previous models (then the authors either modify the existing model(s) or abandon them and propose a new one). Therefore, as a reader, my expectation for this paper is to see a comparison of the newly identified high- and low-velocity anomalies against the predictions from the previous models. For a seismological paper, it is not a good idea to heavily (or even purely) rely on geochemical data to support or refute a model.

Note that it is okay to admit that the new observations cannot fully rule out a model. The authors can still lean towards their preferred model, as long as the evaluation is unbiased. People understand this. However, a paper may lose credibility if the argument shows bias.

Response: Thanks for your comments. Honestly, we would not refute the previous two main models on the late Mesozoic tectonic evolution of the South China Block. In particular, the elegant flat-subduction model has strong influence in the academic community concerning the geology of South China. The sentence in Lines 275-284 (original line numbers) have been revised to avoid further confusion. Actually, we cited the reference of Li and Li (2007) in many places of the main text, especially in places as an additional support (triggering mechanism) to our delamination proposal. Accordingly, the model in Fig. 6a is also modified to include the foundered slab. In our manuscript, we mainly propose an alternative model (delamination) to explain the new geophysical image and geochemical observations in the South China Block. We hope the revised text reads more objectively than the original version.

More specifically, we view the flat slab model as to play an important role in triggering the lithosphere delamination, as stated in lines 280-290. However, we think the flat slab model itself cannot explain all the geological and geophysical observations. For example, we stated in the revised text ("*However, the increasing amount of geochronological data does not show an evident trend of westward younging of early Mesozoic magmatism as a flat-slab model implies (Liu et al., 2020)*"). Based on our newly obtained geophysical images of delaminated/re-laminated lithosphere, we propose that lithosphere delamination is a viable mechanism responsible for the Mesozoic tectonism and magmatism. In the revised text we make it clearer by stating that "*This is not to negate possible slab foundering as suggested earlier (Li and Li, 2007) but to emphasize the linkage between the surface tectonic records and the observed detached high-*V*s bodies in the depth range of 90-150 km below South China (Figs. 3 and 4).*" We also make it clear in the revised text that "*We suggest that the Mesozoic tectonism*

in South China may reflect a composite effect of multiple geodynamic processes including abnormal subduction and lithospheric delamination, and that the Jurassic crustal extension and magmatism mainly resulted from the latter.”

As stated above, we have tried our best to not show any bias against the previous flat slab model. Instead, we have incorporated it to our delamination model. In practice, although the flat slab model could have dominated the tectonics before ~180 Ma, we suggest lithosphere delamination plays a major role in the Mesozoic lithospheric thinning and magmatism in South China. In the revised text, we present the following reasoning:

(Lines 338-356) “ The joint analysis of our high-resolution tomography images (Figs. 2, 3, 4) and geological observations (Figs. 1b, 5) outlines a consistent picture (Fig. 6) of Mesozoic lithospheric delamination and resulting decompression melting below the South China Block. The delamination process started below the Jiangnan orogen during early Jurassic, likely triggered by earlier lithospheric thickening and flat subduction, and subsequently expanded outwards, with the final stage of delamination terminated in the NE during early Cretaceous (Fig. 6a). The dominant effect of lithospheric delamination within most of these regions is supported by multiple observations: (1) the widespread mafic eruption (Fig. 1b) with a clear lithospheric signature requires melting at shallow depth, implying in-situ lithospheric thinning, (2) the present-day lithosphere beneath most of South China is very thin (~70 km), and the mostly likely period the required thinning could have happened is the Mesozoic, prior to which was tectonic thickening during early Paleozoic (Li & Li, 2007; Shu et al., 2021), (3) Mesozoic mantle xenoliths from the central-south part of the Cathaysia block indicate syn-magmatic sub-continental lithospheric thinning to ~80 km at ca. 170 Ma (Liu et al., 2012), (4) recent plate reconstructions (Müller et al., 2019) and present crustal thickness (Fig. S13) suggest little lateral extension within South China since the Triassic, so the present-day thin lithosphere is better explained by vertical thinning (delamination) than horizontal stretching, and (5) the presence of large Vs gradients across the Moho indicates wide-spread lithosphere delamination below South China (Fig. 3f).”

3. On the presentation and interpretation of the results. The authors now show more profiles and depth slices (Figs. 2, 3, S9, S10), allowing me to raise several questions.

(1) In Fig. 2, Vp model, LV1 is centered on 27N, 114E (Hunan-Jiangxi border) at 80 km and 160 km. In Fig. S9, the Vs velocity here is relatively low, but not as low as in many other areas. Why does this discrepancy occur? What does this imply?

Response: LV1 in Fig. 2 mainly reflects the velocity feature for depths around 160 km and deeper. In comparison, the Vs model in Fig. S9 (now Fig. S10) covers the depth range of 90-130 km. In addition, the Vp model in Fig. 2 has lower resolution than the Vs model in Fig. S9 (1.5° versus 0.5°). This is why this discrepancy occurs.

(2) Figure S9 shows depth slices at 90-130 km. How do these depth slices support the statement about the crust (~30 km) in Line 289-292? I don't get it.

Response: Sorry about this mistake. We forgot to update the figure number. It was supposed to be Fig. S10 (now Fig. S13 in the revised version).

(3) Line 158: “Between 114°E and 120°E, a small and faint high-velocity blob exists within the asthenosphere shown by low-velocity anomalies (LV2)”. This sentence is rather confusing. Is the “high-velocity” blob shown by a “low-velocity” anomaly?

Also, I could only see a “small and faint high-velocity blob” at ~120 km depth at 116°E on profile B-B’. Using “Between 114°E and 120°E” to describe its location is not precise at all. Also, if it is important and worthy of mention, please give a name to this small and faint high-velocity blob.

In addition, if this blob is worthy of mention, then how to prove that it is not an artifact?

Response: In the revised version, the related description has been removed to simplify the descriptions for these profiles.

(4) From Line 143, now the authors explain that the V_s gradient is measured in the vertical direction. Then I suggest the authors say more about the geological implications of V_s gradient. In other words, why is it important to calculate V_s gradient? What geological processes are associated with high V_s gradient, and what leads to low V_s gradient? For example, how does magma underplating at the Moho depth (such as those shown in Figure 6) affect the V_s gradient?

Response: Thanks for the suggestion. We have talked more about V_s gradients across the Moho, as follows:

“From the V_s model, V_s gradients across the Moho can be derived from V_s values at nodes immediately above and below the actual Moho interface, which have the unit of 1/s and can be used to depict the current geodynamic status of the Moho. For a stable Moho, V_s gradients across the Moho are expected to be medium. If the lower crust is delaminated, V_s gradients across the Moho will be large because the velocity differences between the juvenile lower crust and mantle are greater. On the other hand, if the Moho is destructed by the upwelling hot materials, V_s gradients across the Moho would be lower.”

(5) Let’s assume that V_s gradient is an indicator for delamination. In Figure 3F, the high V_s gradient zone trends NE-SW, roughly following the Jiangnan orogenic belt. However, in Figure 4, the interpreted delaminated/relaminated lower lithosphere (red isosurface along the east edge of the Yangtze cratonic lithosphere) is trending north-south. These two orientations differ by 60-70 degrees. This is a significant difference. Why?

Response: The high V_s gradient zone in Figure 3F depicts the model around Moho (~35 km). In comparison, the interpreted relaminated lithosphere in Figure 4 is located deeper than ~90 km. Therefore, it is possible the two types of measurements have different orientations.

(6) I suggest the authors make a movie to show the three-dimensional geometry of the high V_s anomalies. It is always challenging to use a piece of paper to show a 3D object. Therefore, a movie always helps. Showing more figures from different angles (in map view, cross-section view, and oblique view) in the supplementary file should also be useful to deliver the message to the readers.

Response: In the supplementary file, we have included additional 8 cross sections of the Vs model (4 along latitude and 4 along longitude, respectively) in Fig. S12.

4. Some problematic (or wrong) statements:

(1) The authors argue that “it is difficult for oceanic subduction that continued from Triassic to Cretaceous to explain” two major observations (Line 324-331). I think Prof. Zheng-Xiang Li’s serial works over the past 2 decades had explained both points listed here, at least in the first order. See for example Li and Li 2007 *Geology* (please, read this paper again, so that you will see why the statement “magmatism *STARTED* in west Cathaysia at 180 Ma” (Line 326-327) is not faithful to the well-known geological record in South China).

See also:

Zheng-Xiang Li et al 2012 *Tectonophysics* <https://doi.org/10.1016/j.tecto.2012.02.011>;

Meng et al 2012 *Lithos* <https://doi.org/10.1016/j.lithos.2011.11.022>;

Xian-Hua Li et al 2012 *CG* <https://doi.org/10.1016/j.chemgeo.2011.10.027>;

Zhu et al 2017 *Lithos* <https://doi.org/10.1016/j.lithos.2017.10.008>;

Cui et al 2021 *GR* <https://doi.org/10.1016/j.gr.2021.06.021>.

Response: Thanks for your comments. We now notice the paper of Li and Li (2007) had explained the initiation of ca. 180 Ma magmatism and the continuous subduction from Triassic to Cretaceous as discussed by the above series of papers. The previous sentence was not well written to clearly show our main intention. Now the sentence has been revised to avoid confusion.

(2) Line 52: It's unclear and unfair to say that "it remains unclear that ..." without detailed explanations. On what issues exactly does the relationship between delamination and other tectonics processes remain unclear? If this is not elaborated on, I can almost foresee that other readers who also work on this topic would feel offended. I don't think it's your intention here to belittle the previous works. So, I suggest rewriting this sentence.

Response: Thanks. It is now revised as “*It remains unclear whether similar delamination events have occurred below other parts of East Asia and how they were related to widespread intracontinental tectonic and magmatic processes.*”

(3) Line 117: The Taiwan island cannot be simply equivalent to “an orogenic belt”. Instead, the island contains many more tectonic units and has experienced complex geologic history (such as the eastward subduction of the Eurasian plate under the Philippine Sea plate under most of the island, and the Luzon arc-continent collision). I am afraid that the authors are not familiar with Taiwan geology. So I suggest, again, that the authors take my previous suggestion and associate LV3 with the Okinawa backarc extension. (The East China Sea rifting is also part of the story here, but I think it can be omitted for now, just to keep things simple.) I could see here that the authors try to associate every low-velocity anomaly with a major fault or a plate boundary. If my speculation is correct, then associating LV3 with Okinawa makes perfect sense because the present-day Eurasia-Philippine Sea plate boundary exists there.

Response: We have deleted “an orogenic belt” in the revised version. Now the sentence is revised as follows: “...and LV3 at the north edge of the Taiwan island, westward extending to the inland (Fig. 2b), which are also imaged in the global tomography model of Fukao and Obayashi (2013) and could be associated with the Okinawa backarc extension.”

(4) Line 154: The logic relation is problematic here. By using “due to”, are you suggesting that the development of LV2 caused the lithosphere near Tanlu to break? Please keep in mind that correlation does not imply causation.

Also, this entire paragraph describes the observations. I suggest not including opinions or interpretations here. “Due to” means an opinion.

Response: “The lithosphere near the Tanlu Fault (TLF; Fig. 1a) seems to be broken due to the development of low-velocity anomalies (LV2)” has been removed in the revised version. And this paragraph is also simplified to not include opinions or interpretations.

(5) In my concern 2(2) in the previous round of review, I asked if a "delaminated" body can still be detected after staying in the convective mantle and being warmed for 100+ Myr. The authors pointed out that a "relaminated" lithosphere, once cooled down, can be detected as a high-velocity anomaly. Their statement is correct on the "relaminated" lithosphere, but I cannot see how this helps to explain my question on the *delaminated* lithosphere.

Response: For the delaminated lithosphere, if they stayed in the hot mantle for over 100 million years, they could be significantly warmed up so as to not appear as high velocity anomalies. In fact, our seismic image may not have adequate resolving to reveal such deep (>400 km depth) delaminated material. However, in this study, the imaged high velocity bodies, which is the main target of study here, are located in the depth range of 90-150 km. These structures are most likely to be the relaminated lithosphere.

5. Other minor issues and suggestions:

Line 56: Reference is needed here to support the Neoproterozoic Yangtze-Cathaysia amalgamation, because Li and Li 2007 Geology does not focus on this topic.

Response: Thanks. We have added a new reference and cited the reference Shu et al. (2021) here.

Line 57: Please start a new sentence at “which”. This sentence is too long now. Long sentences tend to lower readability.

Response: Thanks. We have revised this sentence to two sentences according to your suggestion.

Line 77: remove “remains”.

Response: Removed.

Line 96-97: please remove “extension” and add references (such as Li and Li 2007 Geology) to support this statement.

Response: Thanks. We replace “extension” with “expansion” and cited a reference Liu et al. (2020) here. This reference gives an updated spatio-temporal distribution of the Late Mesozoic felsic rocks.

Figure 2(C): please use white color for the label “Yangtze”. Also, explain what the white dots denote in the figure caption.

Response: “Yangtze” is already in the white color. The white dots denote earthquakes within 1° of the profile. This is now added in the figure caption.

The paragraph that begins at Line 147: This long paragraph describes the seismological observations. However, it is not written concisely. The authors described every profile individually, making it too long to grasp the core observations. It would be okay to write in this way if this manuscript is submitted to JGR. But for a short paper like NC, a better practice is to point out a key anomaly in one and only one sentence and refer to the relevant panel(s) in Figure 3 (and supplementary figures). This way, the readers don’t need to memorize all the details on every profile first to build an imaginary 3D model in their minds.

Response: Following the suggestion, we have significantly simplified the descriptions for these profiles by mainly focusing on high velocity bodies in the asthenosphere. We still want to describe each profile so that the differences among these profiles are clearly presented, which are important for our interpretations in later sections.

Line 245: “progressive” is more accurate than “steady” in this sentence. “A steady increase” may mean an increase at a steady rate over time, which may not be the case here.

Response: Thanks. Replace “steady” with “progressive” according to your suggestion.

In the title and throughout the manuscript, I suggest using “South China” (with an uppercase “S”) instead of “south China”. “South China” in this study refers to the specific continental block. When one wants to refer to the approximate geographic region, “southern China” is more commonly used (which is also more grammatically correct). Please refer to these two Wikipedia pages to get a general idea of the usage of South versus southern.

https://en.wikipedia.org/wiki/Northern_and_southern_China

https://en.wikipedia.org/wiki/South_China_Craton

Response: Thanks. “south China” has been unified to “South China” according to your suggestion.

In the Supplementary file, please check the spelling and grammar again. Some space is missing, such as “Qaidambasin” in Figure S1 caption.

Response: Thanks for your careful reading. We have checked the Supplementary file and corrected some spelling and grammar issues.

In Figure S1, the “Philippine” Sea plate has a double “p” and its subduction polarity is northwestward. Please correct these two errors.

Response: Thanks. These two errors have been corrected.

In Figure S3, please also show the input model for the resolution test. Also, why is there a low-velocity anomaly (around 28N 113E) at the center of the output model at 40 km?

Response: The input model is now included in the revised version. The reason that there is a low-velocity anomaly at the center of the output model is because this area is not well resolved at this depth.

Figure S10 mentions that adakitic rocks are shown by small solid dots. But I could only see cities shown by this symbol. The adakitic rocks are not plotted.

Response: We have removed “The locations of late Mesozoic adakitic rocks are indicated in the two figures as small solid circles” from the caption.

REVIEWER COMMENTS

Reviewer #2 (Remarks to the Author):

NCOMMS-22-36882C

Zhang and co-authors have made another round of revision for their manuscript titled "Lithospheric delamination evidenced from seismic imaging and its controls on the Mesozoic Basin and Range Magmatic Province in South China". There are four major improvements: (1) the newly added explanation of Vs gradient across Moho, (2) the eight sections in Supplementary Figure S12, (3) the separation of the isotopic data based on geography, and (4) a more balanced, unbiased view on the previous models. Overall, the authors did a good job. Below I raise one remaining issue and a suggestion. My recommendation is Minor Revision.

An issue in Figure 4:

Thanks to the newly added Supplementary Figure S12 (especially sections I, II, VI, and VII), finally, I have a better understanding (or imagination) of the delaminated body under South China. In my previous review, I pointed out that the N-S trend of the reddish isosurface (or triangulated irregular network (TIN) annotated as "delaminated/relaminated lower lithosphere") is incompatible with the N60E-trend high Vs gradients across Moho. I therefore questioned the connection between the two.

Now with Figure S12, I speculate that the reddish isosurface in Figure 4 may contain some artifacts. The authors did not explain which Vs value they chose to build this isosurface. I assume the authors chose a fixed Vs value to build this isosurface. If this is correct, in my eyes, such a huge, north-trending isosurface may result from a selected Vs value that is too low. I suggest the authors try some higher Vs values, such as 4.40 or 4.45 km/s, to find the optimal image in which (1) the isosurface of the "delaminated/relaminated body" better matches the region of high Vs gradient across Moho under south-central South China (2) the isosurface of the "delaminated/relaminated body" under South China is disconnected from the high velocity blob under the southern margin of North China. In addition, please explain how this isosurface is generated in the caption.

One suggestion:

I strongly suggest the authors upload the new Vs model to a data repository and share it with the public. As I mentioned previously, the new Vs model is worth publishing (even if I had not been convinced by the geological interpretation). In the long run, a tomography model is useful if and only if the model is made available to the community. For example, the FWEA model by Tao, Grand, and Niu (2018, <https://doi.org/10.1029/2018GC007460>) is a useful model to a lot of people and it has been cited 165 times since 2018. The MIT-P08 model by Li, van der Hilst et al. (2008, <https://doi.org/10.1029/2007GC001806>) has gained 698 citations and the number is still increasing. Same for the S40RTS, LLNL, UU-P07, Sigloch-NAm-2011, and other publicly available models. On the other hand, many other tomography papers may be just as good, but do not have such an impact on the community, partially because the authors didn't make their models available to everyone. Therefore, I suggest the authors join the "open access" trend and make this model publicly available.

RESPONSE TO REVIEWERS' COMMENTS

Reviewer #2 (Remarks to the Author):

NCOMMS-22-36882C

Zhang and co-authors have made another round of revision for their manuscript titled “Lithospheric delamination evidenced from seismic imaging and its controls on the Mesozoic Basin and Range Magmatic Province in South China”. There are four major improvements: (1) the newly added explanation of Vs gradient across Moho, (2) the eight sections in Supplementary Figure S12, (3) the separation of the isotopic data based on geography, and (4) a more balanced, unbiased view on the previous models. Overall, the authors did a good job. Below I raise one remaining issue and a suggestion. My recommendation is Minor Revision.

Response: Thanks very much for your constructive comments that are very helpful for improving our paper. We are also happy to see that our response to the last round of revision addresses well your major concerns. For your comments on Figure 4 and data sharing, please see detailed responses below.

An issue in Figure 4:

Thanks to the newly added Supplementary Figure S12 (especially sections I, II, VI, and VII), finally, I have a better understanding (or imagination) of the delaminated body under South China. In my previous review, I pointed out that the N-S trend of the reddish isosurface (or triangulated irregular network (TIN) annotated as “delaminated/relaminated lower lithosphere”) is incompatible with the N60E-trend high Vs gradients across Moho. I therefore questioned the connection between the two.

Now with Figure S12, I speculate that the reddish isosurface in Figure 4 may contain some artifacts. The authors did not explain which Vs value they chose to build this isosurface. I assume the authors chose a fixed Vs value to build this isosurface. If this is correct, in my eyes, such a huge, north-trending isosurface may result from a selected Vs value that is too low. I suggest the authors try some higher Vs values, such as 4.40 or 4.45 km/s, to find the optimal image in which (1) the isosurface of the “delaminated/relaminated body” better matches the region of high Vs gradient across Moho under south-central South China (2) the isosurface of the “delaminated/relaminated body” under South China is disconnected from the high velocity blob under the southern margin of North China. In addition, please explain how this isosurface is generated in the caption.

Response: The Figure 4 caption stated that “Three-dimensional view of high Vs anomalies (>0.5%) in South China mantle lithosphere. Vs anomalies refer to Vs perturbations with respect to average velocities at each depth. Colors denote depths where high Vs anomalies are located.” This means that we did not choose a fixed Vs value to build the isosurface. Instead, we set an anomaly threshold (0.5% in the current version) to illustrate the geometry of the anomalous lithosphere. The Vs anomaly is calculated with respect to the average velocities at each depth. Following the Reviewer’s additional suggestion, we have tried several different anomaly thresholds of 1.0%, 1.5% and 2.0%. As can be seen in the following figures, the features of interest are almost the same for thresholds of 0.5% and 1.0%. With increased threshold, the size of delaminated/relaminated lower lithosphere becomes smaller, but the N-S trend stays the same. Therefore, the features showing in

Figure 4 are not artifacts due to an inappropriate threshold. As we responded in the last round, we believe because the high-Vs gradient zone (near Moho) and the interpreted relaminated lithosphere (below ~90 km) are located at different depths, it is possible that these two types of measurements have different orientations.

Three-dimensional view of high Vs anomalies (>0.5%)

Three-dimensional view of high Vs anomalies (>1.0%)

Three-dimensional view of high Vs anomalies (>1.5%)

Three-dimensional view of high Vs anomalies (>2.0%)

One suggestion:

I strongly suggest the authors upload the new Vs model to a data repository and share it with the public. As I mentioned previously, the new Vs model is worth publishing (even if I had not been convinced by the geological interpretation). In the long run, a tomography model is useful if and only if the model is made available to the community. For example, the FWEA model by Tao, Grand, and Niu (2018, <https://doi.org/10.1029/2018GC007460>) is a useful model to a lot of people and it has been cited 165 times since 2018. The MIT-P08 model by Li, van der Hilst et al. (2008, <https://doi.org/10.1029/2007GC001806>) has gained 698 citations and the number is still increasing. Same for the S40RTS, LLNL, UU-P07, Sigloch-NAm-2011, and other publicly available models. On the other hand, many other tomography papers may be just as good, but do not have such an impact on the community, partially because the authors didn't make their models available to everyone. Therefore, I suggest the authors join the "open access" trend and make this model publicly available.

Response: We totally agree with your suggestion that the model should be shared. The Vs model produced in this study and assembled geochemistry dataset used in Figure 5 are now shared in a public data repository at <https://doi.org/10.17605/OSF.IO/3PXRK> or <https://osf.io/3pxrk/>

Finally, we would like to acknowledge you again for your constructive and thorough comments through this reviewing process, which greatly helped us to improve the interpretations of our results.

REVIEWERS' COMMENTS

Reviewer #2 (Remarks to the Author):

NCOMMS-22-36882D

Zhang and co-authors have successfully addressed all my concerns. The four figures in the rebuttal letters are self-explanatory. The link to the data repository works. I appreciate the authors' willingness to create more figures to explain and to make their results publicly available. I recommend this manuscript be accepted.

I have some final suggestions, as listed below. They can all be fixed easily in the proof stage, and I don't need to review this manuscript again.

Figure 4: I suggest moving the annotation "Delaminated/relaminated lower lithosphere (>95 km deep)" to somewhere in South China (maybe Guangdong). Placing it in North China could be confusing. Also, I suggest showing two panels (>0.5%, and >1.0% or >1.5% from the rebuttal letter) in the main text. This would help the readers better understand the geometry and location of the delaminated/relaminated body.

Line 45: add 'the' before 'commonly envisaged'

Line 45: change 'for' to 'of': possible cases 'of' lithospheric delamination

Line 74: add 'of': hampers understanding 'of' the relationship

Line 123: change 'significant' to 'significantly'

Line 169: remove 'clearly'

Line 181: add 'the' before 'Sichuan basin'

Line 245: in 'the' western Cathaysia block

Line 254: 'is' should be 'are'

Line 263: two changes: 'new' insight 'into' the tectonic evolution

Line 288: change 'have' to 'has'

Line 292: remove 'clearly'

Line 308: with ... 'remaining' intact

Line 310: remove 'clearly'

Line 358: due to 'the' rollback

Line 367: 'driving force', two words, no hyphen

Line 376: caused by 'the' rollback

RESPONSE TO REVIEWERS' COMMENTS

Reviewer #2 (Remarks to the Author):

NCOMMS-22-36882D

Zhang and co-authors have successfully addressed all my concerns. The four figures in the rebuttal letters are self-explanatory. The link to the data repository works. I appreciate the authors' willingness to create more figures to explain and to make their results publicly available. I recommend this manuscript be accepted.

I have some final suggestions, as listed below. They can all be fixed easily in the proof stage, and I don't need to review this manuscript again.

Response: We are very happy to see our revision is satisfactory and well addressed your comments. We greatly appreciate your comments during this reviewing process, which are very helpful for improve the quality of our paper.

Figure 4: I suggest moving the annotation "Delaminated/relaminated lower lithosphere (>95 km deep)" to somewhere in South China (maybe Guangdong). Placing it in North China could be confusing. Also, I suggest showing two panels (>0.5%, and >1.0% or >1.5% from the rebuttal letter) in the main text. This would help the readers better understand the geometry and location of the delaminated/relaminated body.

Response: Following your suggestion, we have included two panels with Vs anomalies >0.5% and >1.0%. The annotation "Delaminated/relaminated lower lithosphere (>95 km deep)" is also according moved.

Line 45: add 'the' before 'commonly envisaged'

Line 45: change 'for' to 'of': possible cases 'of' lithospheric delamination

Line 74: add 'of': hampers understanding 'of' the relationship

Line 123: change 'significant' to 'significantly'

Line 169: remove 'clearly'

Line 181: add 'the' before 'Sichuan basin'

Line 245: in 'the' western Cathaysia block

Line 254: 'is' should be 'are'

Line 263: two changes: 'new' insight 'into' the tectonic evolution

Line 288: change 'have' to 'has'

Line 292: remove 'clearly'

Line 308: with ... 'remaining' intact

Line 310: remove 'clearly'

Line 358: due to 'the' rollback

Line 367: 'driving force', two words, no hyphen

Line 376: caused by 'the' rollback

Response: These issues have been taken care of in the revision. Thanks for catching them.